# CONTRA: CONFORMAL PREDICTION REGION VIA NORMALIZING FLOW TRANSFORMATION

**Zhenhan Fang, Aixin Tan**
Department of Statistics and Actuarial Science
The University of Iowa
Iowa City, IA 52242, USA
`{zhenhan-fang, aixin-tan}@uiowa.edu`

**Jian Huang**
Department of Applied Mathematics
The Hong Kong Polytechnic University
Hongkong, China
`j.huang@polyu.edu.hk`

## ABSTRACT

Density estimation and reliable prediction regions for outputs are crucial in supervised and unsupervised learning. While conformal prediction effectively generates coverage-guaranteed regions, it struggles with multi-dimensional outputs due to reliance on one-dimensional nonconformity scores. To address this, we introduce CONTRA: CONformal prediction region via normalizing flow TRAnsformation. CONTRA utilizes the latent spaces of normalizing flows to define nonconformity scores based on distances from the center. This allows for the mapping of high-density regions in latent space to sharp prediction regions in the output space, surpassing traditional hyperrectangular or elliptical conformal regions. Further, for scenarios where other predictive models are favored over flow-based models, we extend CONTRA to enhance any such model with a reliable prediction region by training a simple normalizing flow on the residuals. We demonstrate that both CONTRA and its extension maintain guaranteed coverage probability and outperform existing methods in generating accurate prediction regions across various datasets. We conclude that CONTRA is an effective tool for (conditional) density estimation, addressing the under-explored challenge of delivering multi-dimensional prediction regions.

## 1 INTRODUCTION

In unsupervised and supervised learning, density and conditional density estimation are frequently used to communicate the inherent variability in the potential outcome and assess the reliability of estimations and predictions (Hall & Yao, 2005; Guhaniyogi et al., 2014; Dalmasso et al., 2020). These methods can report regions that aim to capture the true outcome with a specified probability, such as 90%, based on the estimated density. However, the actual coverage rate depends on the underlying model assumptions and does not come with any inherent guarantees.

Conformal prediction is a way to produce regions with guaranteed coverage rate of future outcomes non-asymptotically and free of model assumptions (Papadopoulos et al., 2002; Vovk et al., 2005; Lei et al., 2018; Romano et al., 2019; Lei et al., 2015; Izbicki et al., 2020; Chernozhukov et al., 2021). Briefly, the most popular *split conformal* approach partitions the whole dataset into a *proper training set* and a *calibration set*. A predictive model is trained using the former, and non-conformity scores are calculated on the latter that measure the deviation of the predictions from the actual outputs. Quantiles of the non-conformity scores are used to set threshold values, producing *prediction regions* that capture future outcome with desired probabilities. More accurate predictive models usually yield tighter conformal prediction regions.

The conformal prediction literature has primarily focused on one-dimensional output settings, partly because the conformal idea depends on quantiles. Among the relatively few multi-targeting methods in the literature, most restrict region shapes to be boxes or ellipsoids. For complex conditional distributions of the output, such as those with multi-modes or unequal-tails in different directions, boxes or ellipsoids encompass low-density areas to ensure validity, and become inflated. Two recent approaches that allow flexible shape for multi-dimensional conformal regions are the spherically transformed directional quantile regression (ST-DQR) (Feldman et al., 2023) and the probabilistic

conformal prediction (PCP) (Wang et al., 2022). But their prediction regions are composed of numerous balls, which often lead to highly irregular boundaries and more disconnected regions than desirable, hindering interpretability.

We propose CONTRA, a new method that produce prediction regions for multi-dimensional outputs. CONTRA stands for CONformal prediction region via normalizing flow TRAnsformation. Here, normalizing flows (NF) (Dinh et al., 2016; Huang et al., 2018; Chen et al., 2019; Kingma & Dhariwal, 2018; Papamakarios et al., 2021) and conditional normalizing flows (CNF) (Winkler et al., 2019) are generative methods that provide samples, hence density estimates, for the output.

An example is shown in Figure 1 for the prediction region of dropoff locations of Taxi (**y**) given an arbitrary pickup point (**x**, shown as blue pin) in New York city. This is a situation where the conditional density of the outcome is multimodal. Ten different conformal predictions were displayed. The five methods with restricted shapes in (e) and (f) yield large and unnatural prediction regions. The CONTRA region in (b), a variation of CONTRA in (c), and the PCP and ST-DQR methods based on either NF or diffusion models in (d), (g) and (h) produced comparable areas with similar sizes. But both CONTRA prediction regions in (b) and (c) are much more connected than that of PCP and ST-DQR, providing the most effective and interpretable results that maintained theoretical rigor.

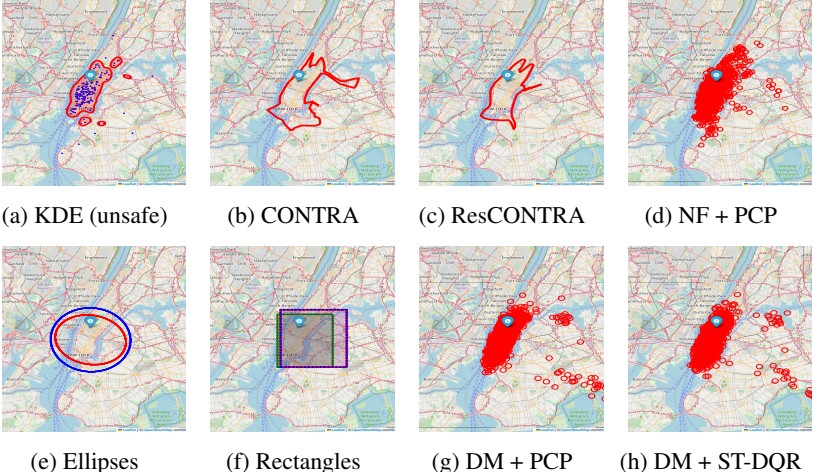

| (a) KDE (unsafe) | (b) CONTRA | (c) ResCONTRA | (d) NF + PCP |
| --- | --- | --- | --- |

| (e) Ellipses | (f) Rectangles | (g) DM + PCP | (h) DM + ST-DQR |
| --- | --- | --- | --- |

Figure 1: NYC Taxi data. Prediction regions of drop-off location given a pickup coordinate (blue pin). (a) shows a 90% high-density region estimated via kernel density estimates (KDE) based on dropoff locations from the 200 nearest pickups to the blue pin. While there is no guarantee of coverage, the shape of this region offers an informal visual reference to help users assess conformal regions. (b)-(f) show various conformal predictions based on NF, trained on 3600 samples and calibrated on 1200. The methods include our proposed CONTRA and ResCONTRA; PCP and ST-DQR (the latter result is not shown due to its similarity to the PCP); NLE and RCP that always lead to elliptical regions; and MCQR, Dist-Slpit, and $CQR_{bon}$ that always lead to rectangular regions; (g) and (h) show the PCP and ST-DQR conformal regions based on a diffusion model, in particular, a Denoising Diffusion Probabilistic Model (Ho et al., 2020) trained with 800 timesteps, learning rate 0.001 and 300 epochs.

We briefly explain here how CONTRA works. Given an input, CNF learns a bijection that maps a latent variable from a simple base distribution, like a standard multivariate Gaussian, to the outcome variable (details in Section 2). After training a CNF on the proper training set, CONTRA maps high-density regions (HDRs) of the base distribution to the outcome, while the split-conformal approach adjusts these regions in the latent space by comparing latent representations of the calibration set to the expected base distribution. Consequently, CONTRA's prediction region is a bijection of a single high-density area, naturally aligning with the output's conditional distribution, circumventing the need to union many regions, as seen in PCP and ST-DQR.

While flow-based models excel in many contexts, there are times when other predictive approaches are preferable. For these situations we invent a variation of CONTRA called ResCONTRA, which enhance any predictive model with a reliable prediction region by training a simple NF on the residuals. The ResCONTRA approach builds on Colombo (2024), where one-dimensional residuals were

transformed to follow base distributions like Uniform or Gaussian using relatively simple bijections. ResCONTRA extends this idea to multi-dimensional outputs and introduces the concept of a symmetric base distribution in $\mathbb{R}^q$ en route to define an ideal non-conformity score. In describing ResCONTRA in section 3.4, we also highlight the need for a three-step calibration procedure to ensure exchangeability that leads to coverage guarantee. Compared to CONTRA, which trains a single model, ResCONTRA seems less efficient as it trains two models on smaller data sets. But ResCONTRA has the potential to excel in situations, say, when $\mathbb{E}(\mathbf{y}|\mathbf{x})$ is highly intricate and can be better learned with techniques like XGBoost rather than an underlying bijection. Some empirical comparisons of the two proposed methods can be found in Section 5 and Appendix G.

We provide details of CONTRA and ResCONTRA in Section 3, review existing multi-output conformal predictions in Section 4. We demonstrate that CONTRA and ResCONTRA achieve the desired coverage probability (e.g., 90%) with high accuracy in Section 5. They outperform shape-restricted methods by providing smaller regions that better reflect the true density. And they have smoother boundaries and improved interpretability compared to the flexible regions of PCP and ST-DQR.

## 2 CONDITIONAL NORMALIZING FLOW

Let $(\mathbf{X}, \mathbf{Y})$ denote a random vector with unknown probability density (or mass) function $p_{\mathbf{XY}}(\mathbf{x}, \mathbf{y})$, supported on $\mathcal{S} \subset \mathbb{R}^{(p+q)}$. For each $\mathbf{x} \in \mathcal{X} \subset \mathbb{R}^p$, the support of $\mathbf{Y}$ is denoted by $\mathcal{Y}_{\mathbf{x}} \subset \mathbb{R}^q$. Suppose a data set $D = \{(\mathbf{x}_i, \mathbf{y}_i)\}_{i=1}^n$ has been drawn from $p_{\mathbf{XY}}(\mathbf{x}, \mathbf{y})$. The objective of a conditional density estimation method is to find an estimate $\hat{p}_{\mathbf{Y}|\mathbf{X}}(\cdot|\mathbf{x})$ of the true conditional density $p_{\mathbf{Y}|\mathbf{X}}(\cdot|\mathbf{x})$ for all $\mathbf{x} \in \mathcal{X}$ based on $D$. In a parametric approach, where candidate models are $\mathcal{P} = \{p_{\mathbf{Y}|\mathbf{X}, \theta}; \theta \in \Theta\}$, the maximum likelihood principle leads to $\hat{p}_{\mathbf{Y}|\mathbf{X}}(\cdot|\mathbf{x}) = p_{\mathbf{Y}|\mathbf{X}, \hat{\theta}}(\cdot|\mathbf{x})$, where $\hat{\theta}$ is given by:

$$\hat{\theta} = \arg \max_\theta \prod_{i=1}^n p_{\mathbf{Y}|\mathbf{X}, \theta}(\mathbf{y}_i|\mathbf{x}_i) = \arg \max_\theta \sum_{i=1}^n \log p_{\mathbf{Y}|\mathbf{X}, \theta}(\mathbf{y}_i|\mathbf{x}_i). \tag{1}$$

The CNF approach specifies $\mathcal{P}$ to be such that, each $p_{\mathbf{Y}|\mathbf{X}, \theta}(\cdot|\mathbf{x})$ is a density of a $q$-dimensional random vector $\mathbf{Y}$ that can be transformed via a differentiable bijection to a simple random vector $\mathbf{Z} \in \mathcal{Z}$, such as a $q$-dimensional Gaussian[1]. We denote this transformation by

$$\mathbf{y} = t_\theta(\mathbf{z}, \mathbf{x})$$

such that for any $\mathbf{x} \in \mathcal{X}$, $t(\cdot, \mathbf{x})$ is a differentiable bijection from $\mathcal{Z}$ to $\mathcal{Y}_{\mathbf{x}}$, with inverse function $t^{-1}(\cdot, \mathbf{x})$. Let $\det$ denote the determinant function. The change-of-variable technique implies:

$$p_{\mathbf{Y}|\mathbf{X}, \theta}(\mathbf{y}|\mathbf{x}) = p_{\mathbf{Z}}(t_\theta^{-1}(\mathbf{y}, \mathbf{x})) \left| \det \frac{\partial t_\theta^{-1}(\mathbf{y}, \mathbf{x})}{\partial \mathbf{y}} \right|.$$

It can be extremely challenging to directly constructing an expressive enough collection of transformations, $\{t_\theta, \theta \in \Theta\}$, to capture the true conditional density, $p_{\mathbf{Y}|\mathbf{X}}(\mathbf{y}|\mathbf{x})$. Fortunately, the compositional nature of bijective functions allows complex transformations to be built from simpler ones: $t_\theta(\cdot, \mathbf{x}) = t_{m,\theta} \circ t_{m-1,\theta} \circ \ldots \circ t_{1,\theta}(\cdot, \mathbf{x})$. Accordingly,

$$\log p_{\mathbf{Y}|\mathbf{X}, \theta}(\mathbf{y}|\mathbf{x}) = \log p_{\mathbf{Z}}(\mathbf{z}) - \sum_{l=1}^m \log \left| \det \frac{\partial t_{l,\theta}}{\partial \mathbf{z}^{(l-1)}} \right|,$$

where $\mathbf{z}_0 = \mathbf{z}$, $\mathbf{z}_l = t_{l,\theta}(\mathbf{z}_{l-1}, \mathbf{x})$ for $l = 1, \ldots, m$, and $\mathbf{z}_m = t_\theta(\mathbf{z}, \mathbf{x}) = \mathbf{y}$. And equation 1 can be rewritten as:

$$\hat{\theta} = \arg \max_\theta \sum_{i=1}^n \left[ \log p_{\mathbf{Z}}(\mathbf{z}_i) - \sum_{l=1}^m \log \left| \det \frac{\partial t_{l,\theta}}{\partial \mathbf{z}_i^{(l-1)}} \right| \right]. \tag{2}$$

From equation 2, it's essential to specify transformations $t_l$ that are easy to evaluate Jacobian determinants for. Several structured approaches, including autoregressive flows (Huang et al., 2018), linear flows (Kingma & Dhariwal, 2018), and residual flows (Chen et al., 2019), facilitate such tractability. In this paper, we employ an affine transformation model known as realNVP (Dinh et al., 2016), which leverages coupling layers (Dinh et al., 2014) to improve computational efficiency. Further details on realNVP are included in Appendix C.

---

[1] The general CNF framework allows the base distribution of $\mathbf{z}$ to depend on $\mathbf{x}$ and $\theta$. We intentionally used one free of $\mathbf{x}$ so that the latent $\mathbf{z}$ at different values of $\mathbf{x}$ are comparable and can be pooled for calibration.

## 3 CONTRA: CONFORMAL REGION VIA NORMALIZING FLOW TRANSFORMATION

Having trained a CNF model, $t_{\hat{\theta}}$, it is tempting to perform naive conditional density estimation as follows. Given any $\mathbf{x}_{n+1}$ of interest, define the naive $(1 - \alpha)$ prediction region to be

$$\tilde{C}_{1-\alpha}(\mathbf{x}_{n+1}) = \{\mathbf{y} : \mathbf{y} = t_{\hat{\theta}}(\mathbf{z}, \mathbf{x}_{n+1}), \mathbf{z} \in B_{1-\alpha}\},$$

where $B_{1-\alpha}$ is the q-dimensional ball that is the highest probability region of size $(1 - \alpha)$ in the latent space for the standard Gaussian. However, the actual coverage rate of this region can vary greatly depending on the quality of $\mathbf{y} = t_{\hat{\theta}}(\mathbf{z}, \mathbf{x}_{n+1}), \mathbf{Z} \sim N_q(\mathbf{0}, \mathbf{I})$ as an approximate sampler for $p_{\mathbf{Y}|\mathbf{X}=\mathbf{x}_{n+1}}$. Approximation errors can be high for $\mathbf{x}_{n+1}$ values not well represented in training. Given $\mathbf{x}_{n+1}$, we would like methods to construct $\hat{C}(\mathbf{x}_{n+1}) \subset \mathbb{R}^q$ that satisfies *the marginal coverage guarantee*:

$$\mathbb{P}[\mathbf{y}_{n+1} \in \hat{C}(\mathbf{x}_{n+1})] \geq 1 - \alpha. \tag{3}$$

### 3.1 CONFORMAL REGIONS: FROM THE LATENT SPACE TO THE OUTPUT SPACE

Two general ideas to achieve equation 3 are full conformal prediction (Vovk et al., 2005) and split conformal prediction (Papadopoulos et al., 2002; Lei et al., 2018; 2015; Angelopoulos & Bates, 2021). Full conformal prediction requires recalculating non-conformity scores across the entire dataset for each new instance. Split conformal prediction partitions the whole dataset into two disjoint sets: the *proper training set*, $D_1 = \{(\mathbf{x}_i, \mathbf{y}_i) : i \in I_1\}$ of size $n_1$; and the *calibration set*, $D_2 = \{(\mathbf{x}_i, \mathbf{y}_i) : i \in I_2\}$ of size $n_2 = n - n_1$. We choose the split approach for its computational efficiency in developing CONTRA.

Recall $t_{\hat{\theta}}(\cdot, \cdot) : \mathcal{S} \to \mathbb{R}^q$ stands for the CNF model with a standard Gaussian base trained from $D_1$. Given any $(\mathbf{x}, \mathbf{y})$, there is a latent representation of the output, which we denote by $\hat{\mathbf{z}} = t_{\hat{\theta}}^{-1}(\mathbf{y}, \mathbf{x})$. Denote the collection of latent representations for $D_2$ by

$$\mathcal{Z}_{\text{cal}} = \{\hat{\mathbf{z}}_i \in \mathbb{R}^q : \hat{\mathbf{z}}_i = t_{\hat{\theta}}^{-1}(\mathbf{y}_i, \mathbf{x}_i), i \in I_2\}. \tag{4}$$

Let $r_{1-\alpha}$ denote the $\lceil (1 - \alpha)(n_2 + 1) \rceil$-th smallest member of $\{\|\hat{\mathbf{z}}_i\|_2, i \in I_2\}$, where $\| \cdot \|_2$ is the Euclidean norm. We define the *conformal ball* of size $(1 - \alpha)$ to be

$$\hat{E} = \{\mathbf{z} \in \mathbb{R}^q : \|\mathbf{z}\| \leq r_{1-\alpha}\}. \tag{5}$$

Then $\hat{E}$ contains at least $(1 - \alpha)100\%$ of the points in $\mathcal{Z}_{\text{cal}}$. By the inflation of quantiles lemma (Romano et al., 2019), assuming the points in $D_2$ and the point to predict, $(\mathbf{x}_{n+1}, \mathbf{y}_{n+1})$, are exchangeable, we have

$$\mathbb{P}(\hat{\mathbf{z}}_{n+1} \in \hat{E}) = \mathbb{P}(\|\hat{\mathbf{z}}_{n+1}\| \leq r_{1-\alpha}) \geq 1 - \alpha.$$

The CONTRA prediction region is defined to be the mapping of $\hat{E}$ in the output space:

$$\hat{C}(\mathbf{x}_{n+1}) = t_{\hat{\theta}}(\hat{E}, \mathbf{x}_{n+1}).$$

The algorithm for producing CONTRA prediction region is summarized as Algorithm 1. Its coverage guarantee is stated in Proposition 2 in Appendix A.1.

---

**Algorithm 1** Conformal Region via Normalizing Flow Transformation (CONTRA)

---

**Input :**
    1: Data $\{(\mathbf{x}_i, \mathbf{y}_i)\}_{i=1}^n \in \mathbb{R}^p \times \mathbb{R}^q$.
    2: Miscoverage level $\alpha \in [0, \ 1]$.
    3: A CNF algorithm $\mathcal{A}$ with a standard Gaussian base distribution.
    4: A point $\mathbf{x}_{n+1}$ that needs a prediction region for its output, $\mathbf{y}_{n+1}$.
**Procedure :**
    1: Randomly split $\{(\mathbf{x}_i, \mathbf{y}_i)\}_{i=1}^n$ into two disjoint sets $D_1$ and $D_2$.
    2: Fit $t_{\hat{\theta}}$ by $\mathcal{A}(D_1)$.
    3: Obtain $\mathcal{Z}_{\text{cal}}$ as in equation 4.
    4: Compute $r_{1-\alpha}$ and define $\hat{E}$ as in equation 5.
    5: Compute $\hat{C}(\mathbf{x}_{n+1}) = t_{\hat{\theta}}(\hat{E}, \mathbf{x}_{n+1})$.
**Output :**
    A prediction region for $\mathbf{y}_{n+1}$ is given by $\hat{C}(\mathbf{x}_{n+1})$.

---

## 3.2 PRACTICAL ASPECTS OF PRESENTING CONTRA OUTPUTS

There is flexibility in implementing steps 4 and 5 of Algorithm 1 and in displaying the prediction region. One way is to get samples inside $\hat{E}$ and map them to the output space. Samples can be random or deterministic, using methods like Monte Carlo, grid points, or Quasi Monte Carlo. Another less costly way is to only sample from the *boundary* of $\hat{E}$, which will map to of $\hat{C}$ in the output space. The validity of this approach hinges on Proposition 3 in Appendix A.2, which states that boundaries of sets are preserved under homomorphisms, including the CNF transformation.

## 3.3 CONNECTEDNESS AND VOLUME OF CONTRA

There is no universal criteria for comparing different prediction regions, provided they have guaranteed coverage. However, smooth boundaries and smaller volumes are generally considered desirable characteristics.

Connectedness. Fewer disconnected sets and smoother boundaries for a prediction region means predictions close to each other are more likely to be classified the same way, either inside or outside of the prediction region. These properties contribute to robust and interpretable inferences in practice. The prediction region $\hat{C}$ of CONTRA is indeed closed and connected due to the following.

**Proposition 1.** *(James, 2000, Chap.3) Suppose $E \subset \mathcal{Z}$ is closed and connected, and $t$ is a homeomorphism. Then $t(E) \subset \mathcal{Y}$ is also closed and connected .*

Volume calculation. Assessing, calibrating, and comparing different prediction regions necessitates a tool to calculate the volume of the regions, which can be challenging for irregular multi-dimensional shapes. We propose an approximation method for the volume of $\hat{C}(\mathbf{x}_{n+1})$, using by-products from the training process of CNF. Note that

$$\mathrm{Vol}\left(\hat{C}(\mathbf{x}_{n+1})\right) = \int_{\hat{E}} \left|\det(J_{t_{\hat{\theta}}}(\mathbf{z}))\right| d\mathbf{z} = \mathrm{Vol}(\hat{E}) \int_{\hat{E}} \left|\det(J_{t_{\hat{\theta}}}(\mathbf{z}))\right| \frac{1}{\mathrm{Vol}(\hat{E})} d\mathbf{z} .$$

The above integral can be approximated using a Monte Carlo estimator based on a random sample, $\{\mathbf{z}_b\}_{b=1}^{B}$, drawn uniformly from $\hat{E}$, with density $1/\mathrm{Vol}(\hat{E})$:

$$\widehat{\mathrm{Vol}}\left(\hat{C}(\mathbf{x}_{n+1})\right) = \mathrm{Vol}(\hat{E}) \frac{1}{B} \sum_{b=1}^{B} \left|\det(J_{t_{\hat{\theta}}}(\mathbf{z}_b))\right| .$$

## 3.4 EXTENSION TO WORK WITH OTHER PREDICTION MODELS: RESCONTRA

The CONTRA method proposed so far restricts the fitted model to be a NF. We now present a variation, ResCONTRA, to enable conformal prediction for any user-chose prediction methods. As explained in the introduction, this is inspired by an idea in Colombo (2024). We split the whole dataset into $D_1$, $D_2$ and $D_3$. Here, $D_1$ is used to train the user-chosen point estimator, $\hat{f}$. Residuals, $\mathbf{r}_i = \mathbf{y}_i - \hat{f}(\mathbf{x}_i)$, for points in the latter two sets are calculated. Then the standard CONTRA is applied to $D_2^* = \{(\mathbf{x}_i, \mathbf{r}_i)\}_{i \in I_2}$ and $D_3^* = \{(\mathbf{x}_i, \mathbf{r}_i)\}_{i \in I_3}$, which serve as the proper training and the calibration set respectively in the split conformal framework. Denote the NF that transforms the residuals in $D_2^*$ to latent $\mathbf{z}$ by $t^{-*}$, and the conformal ball $\hat{E}^*$ is calibrated with $D_3^*$. Finally, the prediction region for a new data point is given by

$$\hat{C}^*(\mathbf{x}_{n+1}) = \hat{f}(\mathbf{x}_{n+1}) + t^*(\hat{E}^*, \mathbf{x}_{n+1}) .$$

It's straightforward to see that marginal coverage is guaranteed, as

$$\mathbb{P}(\mathbf{y}_{n+1} \in \hat{C}^*(\mathbf{x}_{n+1})) = \mathbb{P}(\mathbf{r}_{n+1} \in t^*(\hat{E}^*, \mathbf{x}_{n+1})) \geq 1 - \alpha.$$

Remark: If the distribution of $\mathbf{z}$ were perfectly independent of $\mathbf{x}$, Corollary 2.6 of Colombo (2024) suggests that conditional coverage probability of $\mathbf{y}_{n+1}$ given any $\mathbf{x}$ can be achieved. However, since perfect independence is not achievable in practice with finite data, they also derived in Theorem 2.7 a theoretical bound for the potential reduction in conditional coverage probabilities. This bound

depends on the deviation between the learned distribution of $\mathbf{z}$ and the ideal base distribution that is independent of $\mathbf{x}$. Since both CONTRA and ResCONTRA aim to transform $\mathbf{z}$ to have symmetrical distributions free of $\mathbf{x}$, both methods are expected to approximately achieve the desired conditional coverage similar to those shown in Colombo (2024). It is an on-going work to analyze the conditional coverage probabilities of CONTRA and ResCONTRA both theoretically and empirically, with the challenging but important goal of deriving practically useful bounds.

## 4 RELATED WORK

In contrast to CONTRA, traditional conformal prediction methods construct prediction regions directly in the output space. We review some of them below.

### 4.1 CONFORMAL APPROACHES THAT TARGET MULTI-DIMENSIONAL OUTPUTS

Limited conformal methods in the literature target multi-dimensional outputs directly. An example is the robust conformal prediction (RCP) (Johnstone & Cox, 2021), which employs the global covariance matrix to produce an ellipsoid region. The normalized locally ellipsoid (NLE) (Messoudi et al., 2022) extends RCP by incorporating local covariance matrices, making the region adaptive to $\mathbf{x}$.

Two recent methods, PCP and ST-DQR, are closer to CONTRA as they don't restrict shape of the prediction regions. First, the PCP method was proposed by Wang et al. (2022). Given a generative model for estimating the conditional density, PCP defines the non-conformity score of a data point $(\mathbf{x}_i, \mathbf{y}_i)$ as

$$s_i = \min_{1 \le k \le K} \|\mathbf{y}_i - \hat{\mathbf{y}}_i^k\|,$$

where $\{\hat{\mathbf{y}}_i\}_{k=1}^K$ is a sample of the output generated from the estimated conditional density. Then the prediction region at $\mathbf{x}_{n+1}$ is obtained by generating a sample $\{\hat{\mathbf{y}}_{n+1}^k\}_{k=1}^K$ and form

$$\widehat{C}^{\mathrm{PCP}}(\mathbf{x}_{n+1}) = \bigcup_{k=1}^K \{\mathbf{y} : \|\mathbf{y} - \hat{\mathbf{y}}_{n+1}^k\|_2 \le s_{1-\alpha}\},$$

where $s_{1-\alpha}$ is the $\lceil (1-\alpha)(n_2+1) \rceil$-th smallest member of $\{s_i, i \in I_2\}$. Note that each PCP prediction region is the union of $K$ balls. They have flexible, but often irregular, disconnected shapes, and are sensitive to the choice of $K$ and $\alpha$. This brings challenge to interpreting the regions.

Next, the ST-DQR method was proposed by Feldman et al. (2023). It learns an r-dimensional latent representation of the output, e.g., using the conditional variational auto-encoder (CVAE). The latent variable is encouraged to follow a unimodal distribution, so that methods like the directional quantile regression (DQR) are applicable to form convex probability regions for it. Samples are generated in the output space corresponding to points in the latent region. Calibration and the final prediction region are created similarly to the PCP.

ST-DQR and our CONTRA are similar in that they both depend on latent representations, but differ substantially in the latent representation. CONTRA uses bijection and are often able to learn the latent variable to follow the Gaussian reference distribution rather closely. In contrast, the latent variable in ST-DQR is typically of lower dimension than the output, and its distribution is only coarsely similar to a reference. And this is why additional steps like DQR are needed to form latent probability regions in ST-DQR, but our CONTRA can directly use HDR of the reference Gaussian. When calibrating the regions, ST-DQR (and PCP) had to introduce yet another step: union balls around generated samples, and calibrate the common radius of the balls. Whereas CONTRA directly calibrate radius of the Gaussian HDR. Afterall, these methods have their strengths and weaknesses, but each is a valuable addition to the user's toolkit.

### 4.2 ONE-DIMENSIONAL CONFORMAL APPROACHES AND UPGRADING THEM TO CAPTURE MULTI-DIMENSIONAL OUTPUTS

The literature of conformal predictions for one-dimensional output is rather rich and we list a few state-of-the-art methods. Locally adaptive split conformal prediction (Lei et al., 2018) enhances

reliability by adapting to local data variability, while Dist-split (Izbicki et al., 2020) constructs intervals based on the estimated cumulative density function. However, both methods tend to provide broader intervals when the underlying conditional density is not symmetric. Conformalized quantile regression (CQR) (Romano et al., 2019) offers asymmetric intervals using two conditional quantile estimators, but it remains inefficient for multi-modal distributions. This limitation can be addressed by CD-split (Izbicki et al., 2022), which can generate discontinuous prediction intervals, but may encounter stability issues.

Although these methods are designed for handling one-dimensional output, they can be building blocks for valid prediction regions for multi-dimensional outputs. Bonforonni method (Bonferroni, 1936; Dunn, 1961) is a simple but conservative way to do so. Alternatively, we provide in section 4.2.1 a new method called multi-target conformalized quantile regression (MCQR) that lifts traditional one-dimensional CQR to work for multi-dimensional outputs.

### 4.2.1 MCQR: MULTI-TARGET CONFORMALIZED QUANTILE REGRESSION

As in CQR, we train on $D_1$ a pair of lower and upper quantile estimators, $\hat{Q}_j^l$ and $\hat{Q}_j^u$, for each dimension of the output $\mathbf{y} = (y_1, \cdots, y_q)$. Given a weight vector $\mathbf{w} = (w_{11}, w_{12}, \cdots, w_{q1}, w_{q2})$, the choice of which to be discussed later, we define non-conformity of a point $(\mathbf{x}, \mathbf{y})$ as:

$$s = \max_{j=1,\ldots,q} \left\{ w_{j1}(\hat{Q}_j^l(\mathbf{x}) - y_j), w_{j2}(y_j - \hat{Q}_j^u(\mathbf{x})) \right\},$$

Then we set the prediction region to be

$$\widehat{C}^{\mathrm{MCQR}}(\mathbf{x}_{n+1}) = \left\{ \mathbf{y} : \hat{Q}_j^l(\mathbf{x}_{n+1}) - \frac{s_{1-\alpha}}{w_{j1}} \leq y_j \leq \hat{Q}_j^u(\mathbf{x}_{n+1}) + \frac{s_{1-\alpha}}{w_{j2}}, j = 1, \ldots, q \right\}.$$

For any choice of the weight vector $\mathbf{w} = (w_{11}, w_{12}, \cdots, w_{q1}, w_{q2})$, the MCQR satisfies the marginal coverage guarantee defined in equation 3, which we prove in Appendix A.3.

Finally, since $\widehat{C}^{\mathrm{MCQR}}(\mathbf{x}_{n+1})$ is a box, its volume is simply

$$\mathrm{Vol}(\widehat{C}^{\mathrm{MCQR}}(\mathbf{x}_{n+1})) = \prod_{j=1}^{q} \left[ \hat{Q}_j^u(\mathbf{x}_{n+1}) - \hat{Q}_j^l(\mathbf{x}_{n+1}) + \frac{s_{1-\alpha}}{w_{j2}} + \frac{s_{1-\alpha}}{w_{j1}} \right].$$

Coming back to the problem of how to specify the weight vector $\mathbf{w}$, one solution is to minimize the average volume of prediction regions in the calibration set, that is

$$\hat{\mathbf{w}} = \arg \min_{\mathbf{w}} \frac{1}{n_2} \sum_{i \in I_2} \mathrm{Vol} \left( \widehat{C}^{\mathrm{MCQR}}(\mathbf{x}_i) \right).$$

## 5 EXPERIMENTS

In this section, we systematically compare the performance of the proposed CONTRA and ResCONTRA to that of other conformal prediction methods reviewed in section 4, including PCP, NLE, RCP, MCQR, Dist-split and CQR. The last two were designed for one-dimensional outputs. We employ the Bonferroni approach to produce valid multi-dimensional regions, labeled as Dist-split$_{\mathrm{bon}}$ and CQR$_{\mathrm{bon}}$ respectively. Throughout the experiments, the miscoverage rate is set at $\alpha = 0.1$ for each prediction region, hence a nominal coverage rate of $90\%$.

Experiments are conducted on four synthetic and six real datasets. For each real dataset, input variables $\mathbf{x}$ were standardized before model training. There are many ways to implement CNF, all of which would be compatible with our CONTRA method. Here, we applied RealNVP due to its simplicity and computational efficiency. Specifically, we used 6 to 10 coupling layers for each CNF. Each coupling layer (details in Appendix C) involves two neural networks, each including 2 hidden layers with 512 hidden units and ReLU activation function. Optimization is done with the Adam gradient descent method (Kingma & Ba, 2014) with a learning rate of $1 \times 10^{-3}$ for most of cases. Training epochs are mostly set to be 200. To implement ResCONTRA, we selected support vector regression as the predictive model.

We also studied the impact of underfitting and overfitting on CONTRA regions, as well as the impact of data sizes. Practical guidelines are provided for interesting readers in Appendices E and F.

All models were trained on an A30 GPU with 32 GB of memory. The training time for each CNF varied from 1 to 3 minutes, depending on the datasets and RealNVP structures used.

## 5.1 SYNTHETIC DATA ANALYSIS

We experimented with four setups where $\mathbf{Y}|\mathbf{X} = \mathbf{x}$ are mixture-Gaussian, spiral, moon and ring-shaped, respectively. We present results for the first two cases below while leave the latter two in the Appendix D due to limited space. Data generation details can be found in Appendix B. The sample sizes of the training, calibration and testing sets of the mixture Gaussian and spiral examples are 3375, 1125, and 500, respectively. We further split the training set for ResCONTRA into 60% for a support vector regression model and 40% for the first calibration to train a normalizing flow. For additional simulation studies, see Appendix D.

Table 1 shows the empirical coverage probabilities of each method, averaged over 20 replications, with standard errors in the brackets. Each replication is a different split between the proper training set and the calibration set, while the test set is fixed. All conformal methods achieved the nominal level of 0.9. In both setups, CONTRA and PCP are based on the same CNF and generated the smallest two prediction regions. ResCONTRA delivered comparable results, with discrepancies largely because the "user-chosen" . The other methods produced significantly larger regions.

Table 1: Coverage and volume for 2-dimensional 90% conformal regions in two synthetic datasets. Each table entry is the average of results over 20 random splits, with standard error in the parentheses. The method that achieved the smallest volume is in boldface.

|  | Metric | CONTRA | ResCONTRA | PCP | NLE | RCP | MCQR | Dist-split$_{bon}$ | CQR$_{bon}$ |
|---|---|---|---|---|---|---|---|---|---|
| Mixt. | Coverage | **0.91(0.003)** | 0.90(0.003) | 0.91(0.003) | 0.89(0.003) | 0.90(0.002) | 0.89(0.003) | 0.90(0.002) | 0.90(0.002) |
|  | Volume | **64.92(1.015)** | 90.94(1.727) | 71.32(0.845) | 129.88(10.299) | 135.12(12.378) | 109.51(0.412) | 112.38(0.575) | 112.30(0.563) |
| Spiral | Coverage | 0.91(0.003) | 0.90(0.003) | **0.91(0.002)** | 0.91(0.003) | 0.91(0.002) | 0.91(0.003) | 0.91(0.003) | 0.91(0.002) |
|  | Volume | 23.70(1.622) | 41.73(2.275) | **23.22(1.154)** | 68.97(0.917) | 68.16(0.943) | 64.69(0.199) | 65.82(0.157) | 65.42(0.202) |

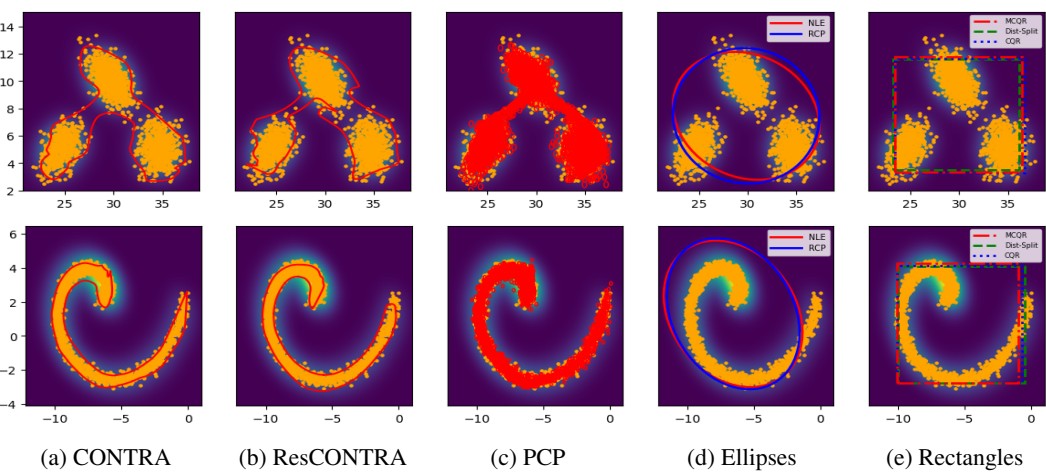

(a) CONTRA     (b) ResCONTRA     (c) PCP     (d) Ellipses     (e) Rectangles

Figure 2: Prediction regions of a two-dimensional outcome given a specific $\mathbf{x}$ value from the test set. Colored lines show the boundary of various prediction regions. In the case of PCP, $K = 3000$ disks are shown. Orange points show a random sample of size 2000 from the true conditional distribution of $\mathbf{y}$ given $\mathbf{x}$.

Beyond numerical metrics, we visualize and compare these conformal regions. Figure 2 shows the prediction regions at an arbitrarily chosen $\mathbf{x}$ value from the test set. CONTRA, ResCONTRA and PCP construct prediction regions that align well with the true HDRs. The remaining four methods are limited to generating elliptical or rectangular regions, hence contain extensive areas of lower density. Among the top three performers, PCP constructs regions from a union of $K$ circles. While increasing $K$ improves the approximation to the true underlying distribution, it also raises computational

costs and creates more complex boundaries. After all, the synthetic data experiments suggest that the proposed CONTRA and ResCONTRA methods outperforms others. They consistently achieve nominal coverage, capture true conditional density with flexible shapes, and maintain smooth, connected regions for robust inference.

## 5.2 REAL DATA ANALYSIS

We next applied CONTRA, ResCONTRA and competing methods to form prediction regions for six datasets from the public-domain. Each dataset was partitioned into 60% training, 20% calibration, and 20% testing. For ResCONTRA, the 60% training portion was further split into 60% for training and 40% for the first calibration. Summaries of their dimensions and sizes are given in Table 2. The first example has a one-dimensional output, and we check if methods for multi-dimensional output work effectively in this special case. Then, we took on examples with two-dimensional output with various input dimensions and training/calibration set sizes. Finally, we tackled a challenging task of density estimation for a four-dimensional output, a significant hurdle for traditional approaches.

Table 2: Summary of dataset structures in real data experiments.

| Dataset | dim($\mathbf{x}$) | dim($\mathbf{y}$) | $n_1$ (training) | $n_2$ (calibration) | $n_3$ (test) |
|---------|------|------|------|------|------|
| Bio | 9 | 1 | 3000 | 1000 | 1000 |
| Taxi | 2 | 2 | 3600 | 1200 | 1200 |
| Energy | 8 | 2 | 460 | 154 | 154 |
| 2D RF | 20 | 2 | 5403 | 1801 | 1801 |
| SCM20D | 61 | 2 | 3000 | 1000 | 1000 |
| 4D RF | 20 | 4 | 5403 | 1801 | 1801 |

Briefly, `Bio` focuses on the physicochemical properties of protein tertiary structures, derived from CASP 5-9 experiment (Rana, 2013). It includes nine predictors that provide information on the structural and geometric properties of molecules, with the aim of predicting the size of the residue. `Taxi` contains longitude and latitude details for pick-up and drop-off location in New York for the year 2016. The goal is to predict the (conditional) distribution of the drop-off location given any pick-up location. This is also the example featured in Figure 1. `Energy` is used to train a predictive model for heating and cooling loads given eight building-related predictors like relative compactness, roof area, surface area and so on (Tsanas & Xifara, 2012). `2D RF` and `4D RF` are based on the same river flow dataset, which comprises more than one year of hourly flow observations from eight sites within the Mississippi River network (Spyromitros-Xioufis et al., 2016). Twenty observations of river network flows at various sites and past time points are used to predict the flows 48 hours into the future at 2 and 4 sites, respectively. `SCM20D` contains 5000 records from the 2010 Trading Agent Competition in Supply Chain Management (Spyromitros-Xioufis et al., 2016). The goal is to predict the mean price 20-days into the future.

Table 3: Coverage and volume for multi-dimensional 90% conformal regions in seven real datasets. Each table entry is the average of results over 20 random splits, with standard error in the parentheses. The method that achieved the smallest volume is in boldface.

| | Metric | CONTRA | ResCONTRA | PCP | NLE | RCP | MCQR | Dist-split$_{bon}$ | CQR$_{bon}$ |
|---|---|---|---|---|---|---|---|---|---|
| Bio | Coverage | 0.90(0.002) | 0.90(0.002) | **0.90(0.003)** | \ | \ | \ | 0.89(0.004) | 0.90(0.002) |
| | Volume | 13.23(0.383) | 12.64(0.178) | **12.54(0.072)** | \ | \ | \ | 13.49(0.132) | 12.55(0.076) |
| Taxi | Coverage | **0.89(0.002)** | 0.89(0.002) | 0.89(0.002) | 0.90(0.001) | 0.90(0.002) | 0.89(0.002) | 0.90(0.002) | 0.91(0.002) |
| | Volume($\times 10^{-3}$) | **8.71(0.290)** | 9.06(0.281) | 8.95(0.132) | 10.77(0.188) | 12.21(0.262) | 10.86(0.283) | 14.65(0.378) | 12.41(0.266) |
| Energy | Coverage | 0.87(0.006) | 0.87(0.009) | **0.88(0.006)** | 0.86(0.006) | 0.85(0.008) | 0.84(0.007) | 0.84(0.010) | 0.87(0.008) |
| | Volume | 18.24(1.269) | 22.76(4.238) | **16.40(0.901)** | 19.14(1.982) | 26.04(2.888) | 25.39(2.238) | 32.12(2.383) | 27.73(2.343) |
| 2D RF | Coverage | **0.91(0.002)** | 0.90(0.002) | 0.91(0.002) | 0.91(0.002) | 0.90(0.002) | 0.91(0.001) | 0.92(0.001) | 0.92(0.002) |
| | Volume | **5.29(0.074)** | 10.34(0.207) | 7.19(0.164) | 15.50(0.307) | 23.85(0.644) | 11.92(0.185) | 12.14(0.187) | 12.30(0.190) |
| SCM20D | Coverage | **0.89(0.001)** | 0.89(0.003) | 0.89(0.002) | 0.90(0.002) | 0.89(0.002) | 0.89(0.002) | 0.90(0.003) | 0.91(0.002) |
| | Volume($\times 10^4$) | **6.30(0.120)** | 8.91(0.207) | 6.97(0.154) | 10.37(0.599) | 6.52(0.153) | 7.10(0.165) | 8.82(0.231) | 8.50(0.206) |
| 4D RF | Coverage | **0.90 (0.003)** | 0.89 (0.002) | 0.89 (0.003) | 0.89 (0.002) | 0.90 (0.002) | 0.89(0.002) | 0.91(0.001) | 0.92(0.002) |
| | Volume($\times 10^2$) | **0.59(0.043)** | 1.54(0.098) | 1.13 (0.057) | 26.10(4.020) | 52.93(11.403) | 10.92(0.597) | 20.40 (1.219) | 12.68(0.665) |

Table 3 reports the empirical coverage probability and volumn of various prediction regions. The coverage rates are generally very accurate around the nominal level 0.9. Slightly lower coverage

rates that range from $0.84$ to $0.88$ are observed across the different methods for the `energy` dataset. This is not surprising given the *marginal* coverage guarantee established for these conformal methods are meant to deliver the nominal coverage when averaged over all possible training and calibration sets. Hence, smaller training and calibration sizes led to a bigger chance of seeing large deviation from the nominal level and relatively low empirical coverage rate out of 20 replications. In addition, for Dist-split$_{\text{bon}}$, a reduction in coverage is observed when the dimension of $\mathbf{x}$ is high. For `Bio`, we see that CONTRA, ResCONTRA and PCP are as good as state-of-the-art methods designed for one-dimensional problems. In multi-dimensional experiments, CONTRA, ResCONTRA and PCP constructed considerably more compact prediction regions than their competitors, with CONTRA keeping the smallest volume with most cases. Overall, the CONTRA-type methods excel in these numerical metrics and offer smoother regions than other top performers, providing a significant advantage.

## 6 Summary and discussion

The main contribution of our work is the proposal of a new multi-dimensional conformal prediction method, CONTRA, that allows reliable conditional density estimation. We demonstrated with various examples that CONTRA surpasses other methods in offering compact prediction regions with flexible shapes and smooth boundaries for easy interpretation.

Alongside CONTRA, we introduce two new methods, ResCONTRA and MCQR, catering to users with their own preferred methods. ResCONTRA extends any user-selected point predictors with valid prediction regions. MCQR extends CQR, a popular one-dimensional conformal prediction method, to create prediction boxes that aim for optimal compactness among box-shaped conformal regions.

Note that most conformal prediction methods are intrinsically one-dimensional: points in the calibration set are projected into a line via a properly defined one-dimensional conformal score. An empirical quantile of these scores serves as the threshold on the line, and is projected back to the output space to form prediction regions. Any choice of the projection will lead to marginal coverage guarantee under the exchangeability condition. Quality projections are key to generating desirable prediction regions, which exhibit properties like small volume, minimal disconnected sets, and smooth boundaries that allow robust and interpretable conclusions in real-world applications. CONTRA is an intuitive and effective approach for defining such projections. It leverages NF and CNF that possess latent representations of the output, and project points to a line based on the density of their latent representation with respect to the standard Gaussian. This reduces to using the distance of the latent representation to the origin in the $q$-dimensional space as the non-conformity score.

One challenge of CONTRA is it requires training a bijection that transforms $q$-dimensional output to its latent representation. Better trained bijection leads to more compact CONTRA regions. In theory, the ground truth bijection always exists (Bogachev et al., 2005). But in practice, more complex distributions with higher $q$ require more data points and increasingly complicated structures in an NF (or CNF) to learn the true bijection well (Durkan et al., 2019). In the case where data size is small and NF (or CNF) is hard to train, one possible solution is to inspect the latent representations of the training and the calibration points and check their distribution against the standard multivariate Gaussian. Patterns of the deviation such as bias in certain directions, correlation among different directions, asymmetry/skewness, kurtosis and so on could potentially be modeled and utilized to adjust the shape of $\hat{E}$ to maintain $90\%$ empirical coverage and be transformed back to obtain adjusted prediction regions in the output space, all while retaining the coverage guarantee.

The current version of CONTRA uses a Gaussian base distribution for the latent representation. This limits its application to model continuous outcomes. Given the wide application of supervised learning with multi-dimensional discrete or mixed-type outcomes, there is an urgent need to perform uncertainty quantification in these settings by developing effective conformal prediction methods. Since there is a wealth of literature on multi-dimensional classification models as well as one-dimensional conformal methods for discrete outcomes, their confluence has the potential to drive rapid advancements in this direction.

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

## A PROOF

### A.1 MARGINAL COVERAGE GUARANTEE OF CONTRA

**Proposition 2.** *(A simple Corollary to the marginal coverage theorem of split conformal methods) Suppose the sample points in $D_2$ and $(\mathbf{x}_{n+1}, \mathbf{y}_{n+1})$ are exchangeable. Then for any CNF model $t_{\hat{\theta}}$, the corresponding conformal ball $\hat{E}$ based on $D_2$ satisfies*

$$\mathbb{P}\left(\mathbf{y}_{n+1} \in t_{\hat{\theta}}(\hat{E}, \mathbf{x}_{n+1})\right) \geq 1 - \alpha.$$

*Further, if $\hat{\mathbf{z}}_i \in \mathcal{Z}_{cal}$ and $\hat{\mathbf{z}}_{n+1}$ are almost surely distinct, the above probability is bounded above by $1 - \alpha + \frac{1}{n_2+1}$.*

*Proof.* Since $t_{\hat{\theta}}$ is trained by CNF, it is a differentiable bijection. Hence

$$\hat{\mathbf{z}}_{n+1} \in \hat{E} \iff t_{\hat{\theta}}(\hat{\mathbf{z}}_{n+1}, \mathbf{x}_{n+1}) \in t_{\hat{\theta}}(\hat{E}, \mathbf{x}_{n+1})$$
$$\iff \mathbf{y}_{n+1} \in t_{\hat{\theta}}(\hat{E}, \mathbf{x}_{n+1}).$$

Therefore

$$\mathbb{P}[\mathbf{y}_{n+1} \in t_{\hat{\theta}}(\hat{E}, \mathbf{x}_{n+1})] = \mathbb{P}[\hat{\mathbf{z}}_{n+1} \in \hat{E}] \geq 1 - \alpha$$

$\square$

### A.2 SET BOUNDARIES REMAIN BOUNDARIES UNDER HOMOMORPHISM

The following result is well-know, but we provide a proof to be self-contained. Many background materials can be found in (James, 2000). Let $\mathcal{U}$ and $\mathcal{V}$ be two topological spaces. (They will be played by $\mathcal{Z}$ and $\mathcal{Y}$ respectively in CNF.)

**Definition 1.** *A function $t : \mathcal{U} \to \mathcal{V}$ is called a homomorphism if $t$ is bijective and continuous, and $t^{-1}$ is continuous.*

**Proposition 3.** *If $t : \mathcal{U} \to \mathcal{V}$ is a homomorphism and $E \subset \mathcal{U}$, then $t(\partial E) = \partial t(E)$, where $\partial E$ is the boundary of $E$.*

*Proof.* The boundary of $E$ in $\mathcal{U}$ can be expressed by $\partial E = \overline{E} \cap \overline{E^c}$, where $\overline{E}$ denotes the closure of $E$ and $E^c$ denotes the complement of $E$.

Since $t$ is a homomorphism, we have

$$t(\overline{E}) = \overline{t(E)}, \;\; t(E^c) = (t(E))^c.$$

Therefore,

$$t(\overline{E^c}) = \overline{t(E^c)} = \overline{(t(E))^c}.$$

Applying $t$ to the boundary of $E$ , we can obtain

$$t(\partial E) = t(\overline{E} \cap \overline{E^c}) = t(\overline{E}) \cap t(\overline{E^c}) = \overline{t(E)} \cap \overline{(t(E))^c} = \partial t(E).$$

Hence, $t(\partial E) = \partial t(E)$. $\square$

### A.3 GUARANTEED COVERAGE PROOF OF MCQR

**Proposition 4.** *Suppose $(\mathbf{x}_i, \mathbf{y}_i)$, $i = 1, \ldots, n + 1$, are exchangeable, then the prediction region $\widehat{C}^{MCQR}(\mathbf{x}_{n+1})$ constructed by the MCQR algorithm satisfies*

$$\mathbb{P}\left[\mathbf{y}_{n+1} \in \widehat{C}^{MCQR}(\mathbf{x}_{n+1}) | (\mathbf{x}_i, \mathbf{y}_i), i \in I_1\right] \geq 1 - \alpha. \tag{6}$$

*If the non-conformity scores, $s_i, i = 1, \cdots, n + 1$, are almost surely distinct, then the probability in the left hand side of equation 6 is also bounded above by $1 - \alpha + \frac{1}{n_2+1}$.*

*Proof.* First, we show that $\mathbf{y} \in \widehat{C}^{\text{MCQR}}(\mathbf{x})$ is equivalent to $s \leq s_{1-\alpha}$.

$$s \leq s_{1-\alpha}$$
$$\iff \max_{j=1,\ldots,q} \left\{ w_{j1} \left( \hat{Q}_j^l(\mathbf{x}) - y_j \right), w_{j2} \left( y_j - \hat{Q}_j^u(\mathbf{x}) \right) \right\} \leq s_{1-\alpha}$$
$$\iff \left\{ \left( \hat{Q}_j^l(\mathbf{x}) - y_j \right) \leq \frac{s_{1-\alpha}}{w_{j1}} \wedge \left( y_j - \hat{Q}_j^u(\mathbf{x}) \right) \leq \frac{s_{1-\alpha}}{w_{j2}} \text{ for } j = 1, \ldots, q \right\}$$
$$\iff \left\{ \hat{Q}_j^l(\mathbf{x}) - \frac{s_{1-\alpha}}{w_{j1}} \leq y_j \leq \hat{Q}_j^u(\mathbf{x}) + \frac{s_{1-\alpha}}{w_{j2}} \text{ for } j = 1, \ldots, q \right\}.$$

Since non-conformity scores $s_i, i \in I_2$ and $s_{n+1}$ are exchangeable. Applying Lemma 1,

$$\mathbb{P}(s_{n+1} \leq s_{1-\alpha} | (\mathbf{x}_i, \mathbf{y}_i) : i \in I_1) \geq 1 - \alpha.$$

If calibration residuals $s_i$ and $s_{n+1}$ are almost surely distinct,

$$\mathbb{P}(s_{n+1} \leq s_{1-\alpha} | (\mathbf{x}_i, \mathbf{y}_i) : i \in I_1) \leq 1 - \alpha + \frac{1}{n_2 + 1}.$$

Taking expectation over $D_1$, the marginal coverage is guaranteed. $\qquad\square$

**Lemma 1** (Inflation of quantiles, Romano et al. (2019)). *Suppose* $\mathbf{Z}_1, \ldots, \mathbf{Z}_{n+1}$ *are exchangeable random variables. For* $\alpha \in (0, 1)$,

$$\mathbb{P}\{\mathbf{Z}_{n+1} \leq \mathbf{Z}_{(\lceil \alpha(n+1) \rceil, n)}\} \geq \alpha,$$

*Where* $\mathbf{Z}_{(\lceil \alpha(n+1) \rceil, n)}$ *is* $\lceil \alpha(n+1) \rceil$ *smallest value in* $\mathbf{Z}_1, \ldots, \mathbf{Z}_n$, *or* $\alpha$-*th empeical quantile of* $\mathbf{Z}_1, \ldots, \mathbf{Z}_n$. *Moreover, if the random variables* $\mathbf{Z}_1, \ldots, \mathbf{Z}_{n+1}$ *are almost surely distinct, then also*

$$\mathbb{P}\{\mathbf{Z}_{n+1} \leq \mathbf{Z}_{(\lceil \alpha(n+1) \rceil, n)}\} \leq \alpha + \frac{1}{n}.$$

# B    SYNTHETIC DATA STRUCTURE

1. A model with a mixture Gaussian error term.

$$\begin{cases} Y_1 = 3X_1^3 X_2 - 5X_2^2 + 4X_1 X_2 - 6X_2 + 7 + \varepsilon_1 \\ Y_2 = X_1 X_2 - X_2^3 + 3X_1 X_2^2 + 8 + \varepsilon_2 \end{cases},$$

where $\mathbf{Y} = [Y_1, \ Y_2]^T$, $\mathbf{X} = [X_1, \ X_2]^T \sim N(\boldsymbol{\mu}, \mathbf{I}_2)$, $\boldsymbol{\mu} = [-2.0, -1.5]^T$, and $\boldsymbol{\varepsilon} = [\varepsilon_1, \varepsilon_2]^T \sim 0.3N\left([0,0]^T, 0.5(\mathbf{I}_2 + \mathbf{J}_2)\right) + 0.4N\left([5,5]^T, 1.5(\mathbf{I}_2 - \mathbf{J}_2)\right) + 0.3N\left([10,0]^T, \mathbf{I}_2\right)$, $\mathbf{J}_2$ is a 2 by 2 matrix with all elements equal to one.

2. A model with a spiral curve error term.

$$\begin{cases} Y_1 = 2X_1^3 - 3X_2^2 + 5X_2 + X_1 X_2 + \varepsilon_1 \\ Y_2 = X_1^2 X_2 - 4X_2^2 + 3X_1^2 X_2 + 7 + \varepsilon_2 \end{cases},$$

where $\mathbf{X}$ has the same structure as in Model 1; $\varepsilon_1 \sim N(\theta \cos(\theta), 0.2^2), \varepsilon_2 \sim N(\theta \sin(\theta), 0.1^2)$, where $\theta \in (0, 2\pi)$.

3. A model with a moon curve error term. The specification of $\mathbf{Y}$ and $\mathbf{X}$ are the same as the previous setup, And the error term follows a moon-shaped distribution, $\varepsilon_1 \sim N(\cos(\theta), 0.1^2), \varepsilon_2 \sim N(\sin(\theta), 0.1^2)$, $\theta \in (0, \pi)$.

4. A model with a ring error term. The specification of $\mathbf{Y}$ and $\mathbf{X}$ are the same as the previous setup, $\varepsilon_1 = r \cos(\theta), \varepsilon_2 = r \sin(\theta)$, where $r^2 \sim U(r_{\text{inner}}^2, r_{\text{outer}}^2), \theta \sim U(0, 2\pi)$.

## C    Coupling Layer

For a vector $\mathbf{y} \in \mathbb{R}^q$, partition $\mathbf{y}$ into two subspaces: $(\mathbf{y}_{I_1}, \mathbf{y}_{I_2}) \in \mathbb{R}^{q_1} \times \mathbb{R}^{q-q_1}$ and a bijection function $g(\cdot) : \mathbb{R}^{q-q_1} \to \mathbb{R}^{q-q_1}$. Define

$$\begin{cases} \mathbf{z}_{I_1} = \mathbf{y}_{I_1} \\ \mathbf{z}_{I_2} = g(\mathbf{y}_{I_2}; m(\mathbf{y}_{I_1}, \mathbf{x})) \end{cases},$$

In particular, let $g(\mathbf{y}_{I_2}; m(\mathbf{y}_{I_1}, \mathbf{x})) = \mathbf{y}_{I_2} \odot \exp(u(\mathbf{y}_{I_1}, \mathbf{x})) + v(\mathbf{y}_{I_1}, \mathbf{x})$ (Winkler et al., 2019), then the determinant of Jacobian is $\exp(\sum_{j=1}^{q-q_1} u(\mathbf{y}_{I_1}, \mathbf{x})_j)$.

The transformation $t$ is composed by multiple coupling layers. To prevent the composition of two consecutive coupling layers from reducing to the identity function, we can switch the roles of the two subsets (Dinh et al., 2016). For instance, consider a scenario where we aim to transform $\mathbf{y}$ into $\mathbf{z}$ via $\mathbf{w}$. In this case, we have:

$$\textcircled{1} \begin{cases} \mathbf{w}_{I_1} = \mathbf{y}_{I_1} \\ \mathbf{w}_{I_2} = g^{(1)}(\mathbf{y}_{I_2}; m(\mathbf{y}_{I_1}, \mathbf{x})) \end{cases} \qquad \textcircled{2} \begin{cases} \mathbf{z}_{I_1} = g^{(2)}(\mathbf{w}_{I_1}; m(\mathbf{w}_{I_2}, \mathbf{x})) \\ \mathbf{z}_{I_2} = \mathbf{w}_{I_2} \end{cases}$$

## D    Additional Simulation Studies

Table 4: Coverage and volume for 2-dimensional 90% conformal regions in two synthetic datasets. Each table entry is the average of results over 20 random splits, with standard error in the parentheses. The method that achieved the smallest volume is in boldface.

| | Metric | CONTRA | ResCONTRA | PCP | NLE | RCP | MCQR | Dist-split$_{\text{bon}}$ | CQR$_{\text{bon}}$ |
|---|---|---|---|---|---|---|---|---|---|
| Moon | Coverage | **0.91(0.002)** | 0.90(0.003) | 0.91(0.003) | 0.91(0.003) | 0.91(0.003) | 0.90(0.003) | 0.91(0.003) | 0.91(0.003) |
| | Volume | **1.56(0.020)** | 2.65(0.062) | 1.66(0.015) | 3.32(0.040) | 3.28(0.037) | 2.50(0.014) | 2.63(0.012) | 2.62(0.013) |
| Ring | Coverage | 0.90(0.004) | 0.91(0.002) | **0.90(0.002)** | 0.91(0.003) | 0.91(0.003) | 0.89(0.004) | 0.89(0.003) | 0.89(0.003) |
| | Volume ($\times 10^2$) | 3.00(0.217) | 4.57(0.405) | **1.97(0.030)** | 5.53(0.875) | 6.14(1.537) | 5.17(0.009) | 5.19(0.010) | 5.19(0.009) |

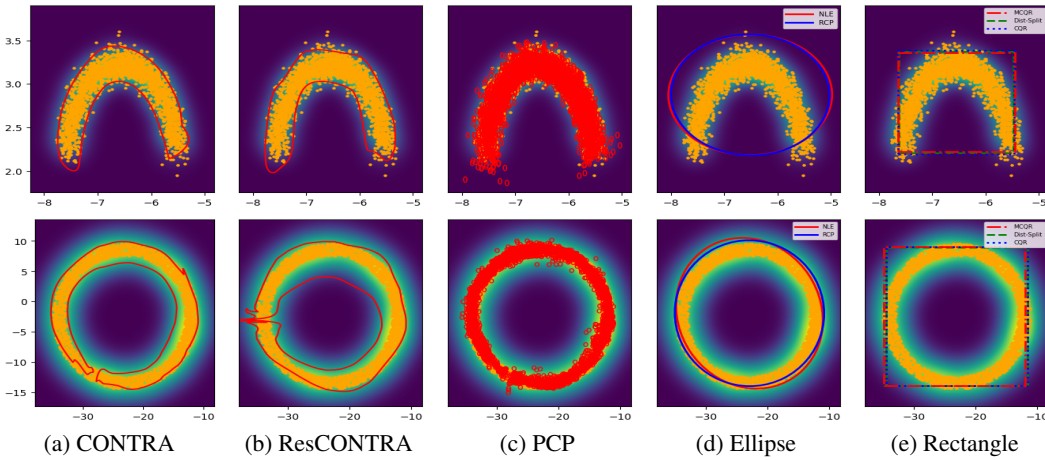

   (a) CONTRA       (b) ResCONTRA      (c) PCP       (d) Ellipse       (e) Rectangle

Figure 3: Prediction regions of a two-dimensional outcome given a specific $\mathbf{x}$ value from the test set. Colored lines show the boundary of various prediction regions. In the case of PCP, $K = 3000$ disks are shown. Orange points show a random sample of size 2000 from the true conditional distribution of $\mathbf{y}$ given $\mathbf{x}$.

## E    The impact of underfitting and overfitting on CONTRA

Using a Normalizing Flow model that is either too complex or too simple can result in conformal regions that are overly sensitive or excessively large. A straightforward guideline to mitigate these overfitting and underfitting issues is to evaluate how closely the latent variables $\mathbf{z}$ from the calibration

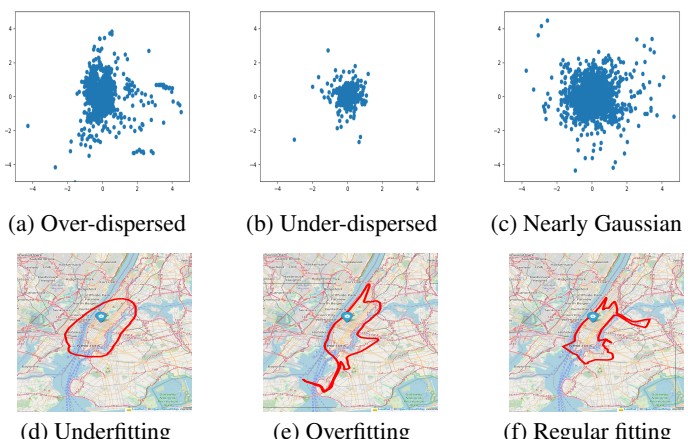

(a) Over-dispersed     (b) Under-dispersed     (c) Nearly Gaussian

(d) Underfitting     (e) Overfitting     (f) Regular fitting

Figure 4: CONTRA prediction regions and latent $\mathbf{z}$ for the calibration set under three different NF models, for drop-off location given a specific pickup coordinate (blue pin) for the NYC taxi data. Data size equals 4800, with a 75%-25% training-calibration split. (a) and (d): An underfitting NF with 2 coupling layers and 16 hidden units per layer, trained for 50 epochs in 4 seconds. (b) and (e): An overfitting NF with 16 coupling layers and 1024 hidden units, trained for 500 epochs in 205 seconds. (c) and (f): A regular fitting NF with 6 coupling layers and 256 hidden units, trained for 200 epochs in 17 seconds.

set resemble a random sample from the standard Gaussian. The top row of Figure 4 provides a visual check for bias, overdispersion, or underdispersion, from which users can tell that the trained model corresponding to the rightmost picture had the best out-of-sample performance among the three, and would be the best choice for constructing CONTRA regions. This is confirmed in the bottom row of Figure 4 . In addition to the visual check, classical metrics can be used to quantify the deviation between the $\mathbf{z}$ sample and the standard Gaussian, aiding in model tuning.

## F  IMPACT OF DATA SIZE ON CONTRA AND ITS MAIN COMPETITOR

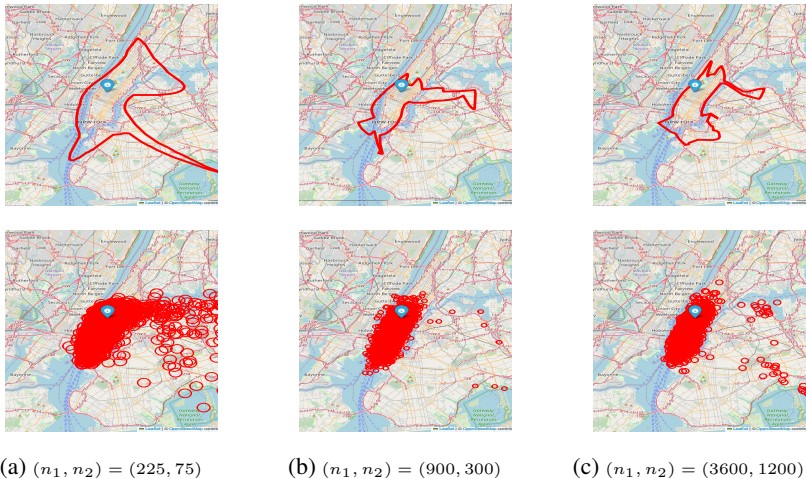

(a) $(n_1, n_2) = (225, 75)$     (b) $(n_1, n_2) = (900, 300)$     (c) $(n_1, n_2) = (3600, 1200)$

Figure 5: The impact of data size on prediction regions of drop-off location given a specific pickup coordinate (blue pin) for the NYC taxi data. The top row shows results of our proposed CONTRA method; the bottom row shows results of ST-DQR based on a diffusion model. Data sizes used for each column are 300, 1200, and 4800, respectively, with a 75%-25% training-calibration split.

We examined the performance of CONTRA and the ST-DQR based on the diffusion model across various sample sizes, with a particular focus on smaller sample sizes that pose challenges for both NF and diffusion model training. The resulting prediction regions for both methods were consistent with

expectations and appeared reasonable. As the sample size increased from very small to moderate, uncertainty decreased, leading to smaller prediction regions. However, as the sample size continued to grow, the prediction region stabilized, remaining approximately the same size, which correctly reflects the inherent uncertainty in predicting outcomes for new subjects.

## G    A COMPARISON BETWEEN CONTRA AND RESCONTRA

This section compares the two proposed methods, CONTRA and ResCONTRA. We use empirical examples to confirm the following heuristics: When NF is effective at learning $\mathbf{y}|\mathbf{x}$, the one-step CONTRA method generally outperforms the two-step ResCONTRA due to its larger training set. Whereas when the relationship between $\mathbf{y}$ and $\mathbf{x}$, such as $\mathbb{E}(\mathbf{y}|\mathbf{x}) = f(\mathbf{x})$, is highly complex but the residual $\mathbf{y} - f(\mathbf{x})$ follows a relatively simple distribution, ResCONTRA tends to provide better prediction regions. This is because ResCONTRA divides the original training data into two subsets. One subset is used to train an estimator for $f(\mathbf{x})$, using any preferred method that specializes for this task of *point estimation*. The second subset is then used to model the distribution of the residual using a relatively simple NF.

Above, we provided a general guideline for choosing between CONTRA and ResCONTRA. In practice, users don't need to decide in advance. They can try both methods and use the method that lead to the prediction region that better suits their needs.

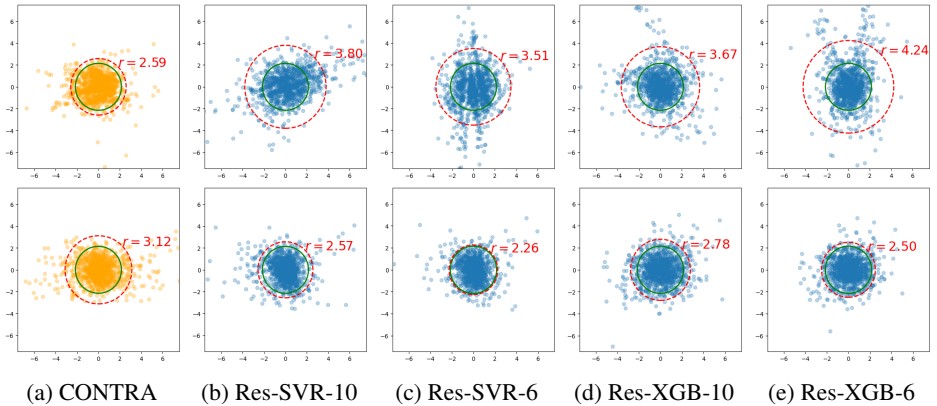

|  | (a) CONTRA | (b) Res-SVR-10 | (c) Res-SVR-6 | (d) Res-XGB-10 | (e) Res-XGB-6 |

Figure 6: Comparing CONTRA to ResCONTRA in terms of the latent variable, $\mathbf{z}$, of the calibration set on two different datasets. Plots of $\mathbf{z}$ that closely resemble the standard bivariate Gaussian distribution result in smaller calibrated radii, $r_{.9}$, and better conformal prediction regions. Five conformal prediction methods are implemented: (a) CONTRA with a 10-layer NF; (b) and (c) ResCONTRA with SVR followed by 10- and 6-layer NFs, respectively; (d) and (e) ResCONTRA with XGBoost followed by 10- and 6-layer NFs, respectively. For CONTRA, the sizes for training, calibration, and testing are (3375, 1125, 500). For ResCONTRA, the training set was further split into 60% to train a point estimator and 40% to train a NF.

|  | CONTRA | SVR-10 | SVR-6 | XGB-10 | XGB-6 |
|---|---|---|---|---|---|
| Mixture-Gaussian (%) | 0.71 | 2.67 | 1.69 | 2.67 | 3.38 |
| Multiplicative Gaussian (%) | 1.24 | 0.27 | 0.09 | 0.00 | 0.00 |

Table 5: The percentage of points out of the $[-7.5, 7.5] \times [-7.5, 7.5]$ range.

For illustration, we apply CONTRA and four different implmentations of ResCONTRA to two examples. The first example is the same mixture-Gaussian example in Section 5.1. And the second example specifies a more complex relationship between $\mathbf{y}$ and $\mathbf{x}$, described in equation 7. For each method that learns the conditional distribution of $\mathbf{y}$ given $\mathbf{x}$, we examine values of the latent variable, $\mathbf{z}$, of the calibration set. Resemblance of the distribution of $\mathbf{z}$ to the bivariate standard normal indicates good out-of-sample learning of the distribution of $\mathbf{y}|\mathbf{x}$. Figure 6 displays the values of $\mathbf{z}$ for the two examples in the top and bottom rows, respectively. All 10 plots show the $[-7.5, 7.5] \times [-7.5, 7.5]$ region of the latent space, while Table 5 shows the proportions of $\mathbf{z}$ falling outside the shown range. Overlaying on each plot is the theoretical 90% highest density region of the bivariate standard Gaussin

(in green) and the smallest circle centered at the origin that contains $90\%$ of the $\hat{\mathbf{z}}$ of the calibration set, with radius denoted by $r$. From the first row of Figure 6 and Table 5, we can see that CONTRA (a) is the best performer for the example with relatively simple model and more complicated error; and from the second row, we can see that ResCONTRA methods, especially (c) and (e) that combine tools specialized for point estimation and relatively simple NF for the residuals, are the best performers to capture the complex relationship between $\mathbf{y}$ and $\mathbf{x}$ while providing small $r$ that leads to small conformal regions.

**Details of the relatively complex dataset**.

This is the dataset that corresponds to the results in the bottom row of Figure 6 and Table 5.

$$\begin{cases} Y_1 = 2X_1^2 e_1 e_2 - 3X_2 + 0.5X_3^3 + X_4 X_5 e_2 - 1.5X_6^2 + 0.7X_7 X_8^2 - 0.3X_9 e_1 + \sin(X_{10}) + 5 \\ Y_2 = -X_1^3 + 4X_2^2 - X_3 X_4 e_2 + 0.8X_5^2 - 2X_6 X_7 e_1 e_2 + 0.6X_8 - 1.2X_9^3 e_1^2 + \cos(X_{10}) + 7 \end{cases},$$
$$(7)$$

where $\mathbf{Y} = [Y_1, \ Y_2]^T$, $\mathbf{X} \sim N(\boldsymbol{\mu}, \mathbf{I}_{10})$, $\boldsymbol{\mu}$ is a 10-dimensional mean vector with each component independently drawn from the Uniform$[-10, 10]$ distribution, and $\boldsymbol{\varepsilon} = [\varepsilon_1, \varepsilon_2]^T \sim N\left(\mathbf{0}, \mathbf{I}_2\right)$.

## H    AGREEMENT BETWEEN CONTRA PREDICTION REGIONS AND HPDS

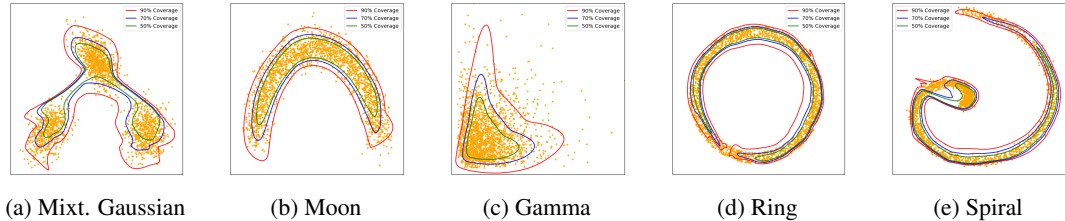

(a) Mixt. Gaussian      (b) Moon      (c) Gamma      (d) Ring      (e) Spiral

Figure 7: CONTRA conformal regions of the output variable $\mathbf{y}$ given some fixed $\mathbf{x}$ value for five setups with coverage levels $50\%$, $70\%$ and $90\%$, respectively. The orange points are random samples of size 2000 from each of the true conditional distribution of $\mathbf{y}$ given $\mathbf{x}$. We can see the CONTRA regions of various levels properly capture the high-density regions in each case.

