# OpenReview forum: "CONTRA: Conformal Prediction Region via Normalizing Flow Transformation"
_ICLR.cc/2025/Conference — ICLR 2025 Poster_

### Official Review · Reviewer_bEdm · 2024-10-28

**Soundness:** 3
**Presentation:** 3
**Contribution:** 2
**Rating:** 6
**Confidence:** 3

**Summary:**

The authors propose a technique to compute valid prediction regions in multiple-output regression tasks. The approach is based on learning a Normalizing Flow that transforms the output conditional probability into a 'simpler' base distribution. The invertibility of the flow allows mapping valid convex intervals of the feature space, e.g. rectangles, to non-convex smooth regions in the original space.

**Strengths:**

- Using Normalizing Flow to improve Conformal Prediction is intriguing and flexible.
- Normalizing Flows and their likes are increasingly popular for training generative models. The proposed method could be further improved by leveraging advances in that domain.
- Multiple-output Conformal Prediction is an important but often overlooked problem.

**Weaknesses:**

- The use of Normalizing Flow is not theoretically justified. The introduction should explain the differences compared to learning more standard feature maps.
- The authors should describe in detail the overlaps between their work and Colombo 2024, which is not mentioned with other related work but seems closely related to the proposed approach.
- It is unclear why the proposed approach is superior to existing multiple-output Conformal Prediction approaches.
- The authors should give more details about the invertible flow, $t_\theta$, and discuss possible overfitting or underfitting problems. Especially in comparison with other non-invertible probability maps (e.g.diffusion models).

**Questions:**

- What is the difference between the proposed approach and methods that learn a feature space map before applying CP, e.g. STDQR?
- What is the difference between using a normalizing flow as a prediction model and a model for the residuals? From Figure 1, the latter seems to produce smoother regions. Is this always the case?
- Can the method be extended to tasks with many more outputs, e.g. images?
- In practice, requiring the map to be invertible may make Normalizing Flow hard to train or less flexible than unconstrained models. Have you tried Normalizing Flow with different structures?

---

> ### Author Response · Authors · 2024-11-22
>
> **Weakness 1**.
>
> The reviewer mentioned "the use of Normalizing flow is not theoretically justified". To respond to this, we would reiterate that CONTRA (the first step involves NF) is theoretically justified in the sense that the resulting prediction region has the marginal coverage property stated in Theorem 1.
>
> We wonder what additionally theoretical result or justification the reviewer is thinking of. It's true that we do not have theoretical proof that the proposed method is optimal in some sense. Instead,  we resort to empirical evidence to show our prediction regions are the most smooth as well as having the smallest or very close to the smallest volume. In addition, there is no single criteria to measure goodness of prediction regions, so it was not the goal of this paper to develop optimal result for any single criteria. Our goal is to provide a practical tool that have coverage guarantee.
>
> **Weakness 2**.
>
> On the relationship of our proposals to Colombo (2024). First, our main method CONTRA is not directly related to Colombo, but it's variation ResCONTRA has a lot of similarity with their work. ResCONTRA is actually a method we very recently added to the paper based on another reader's comments on if the main idea in CONTRA can help users who already have their preferred point prediction methods other than NF. So we got inspired by Colombo and like them, made an attempt to transform the residuals of the calibration set using NF. We realized this week that to ensure marginal coverage guarantee eq(3), we need the exchangeability condition of the calibration set and the future data points, which can be achieved in ResCONTRA only after a second round of calibration, that was not used to train either the user's chosen estimator nor the NF for the residuals. So, we have added that procedure to ensure the theoretical validity of ResCONTRA, regenerated our plots and tables and modified section 3.4 accordingly. In contrast to our two-step calibration, Colombo stick with one-step conformal, gave up marginal coverage, but resort to a theoretical result that gives a lower bound of the coverage probability that take into account the possible deviation of the NF learned distribution from the truth (their Theorem 2.7). So, in comparison to Colombo, our ResCONTRA has the advantage of guaranteed coverage probability, at the price of smaller training size because the original proper training set + calibration set needs a further split into a training and two calibration sets.

---

> > ### Author Response · Authors · 2024-11-22
> >
> > **Weakness 3**.
> >
> > We categorize existing multi-output conformal methods into three major types. The proposed CONTRA is superior than the first two because CONTRA does not restrict its region's shape to elipsoid or boxes, avoiding the region to cover low density areas hence achieving small volume. The proposed CONTRA have similar volume to the third type, PCP and ST-DQR. As all these methods have flexible shapes. Many people will find the connectedness of the CONTRA region more appealing and easier to interpret than highly disconnected regions produced by PCP and STDQR.
> >
> > Here is an additional relevant note on how ResCONTRA is different and hence superior than the Ellipoids. Given point estimator of the output, if the nonconformity scores are defined to be Euclidean distance between the estimates and the observed (residuals) of the calibration set, one would end up with ball-shaped conformal regions. Existing literature used variations of the score defined as distance of the affine or locally affine transformation of residuals \cite{}. These results in conformal regions that are Ellipsoids as shown in Figure 1 (d). In contrast, our ResCONTRA went one step further in allowing any differentiable transformation of the residual when standardizing them to z follow closely the Gaussian distribution. And then define non-conformity scores on the z, which is the most reasonable and hence the superiority of the transformed back conformal region than the ellipsoid ones.
> >
> > Below are more details of what I mean by the three types of existing multi-output conformal methods. The first type helps bring coverage guarantee to a given point estimator of the output, and nonconformity scores are Euclidean distance between the estimates and the observed, with variations of the score defined as distance of the affine or locally affine transformation of residuals. These results in conformal regions that are Ellipsoids as shown in Figure 1 (d). The second type create conformal intervals for each dimension separately, then combine them to make boxes while using Bonferroni type of ideas to ensure overall coverage, as in Figure 1(e). The third type of methods are based on generative method Figure 1(c) that have more flexible shapes, but these regions are always unions of smaller base shapes such as balls, which leaves room for improvement in our opinions.
> >
> > **Weakness 4**.
> >
> > Great question! We ran new simulations and added an Appendix E to the paper to study the impact of the complexity of NF. We also followed your suggestion to run experiments with alternative methods that employ non-invertible maps, specifically, diffusion models. We actually realized NF to be a practical method because it allows a free, simple, and practical diagnostic of whether the model is underfitting, overfitting, or just right. We quote a paragraph from Appendices E
> >         "A straightforward guideline to mitigate these overfitting and underfitting issues is to evaluate how closely the latent variables $\mathbf{z}$ from the calibration set resemble a random sample from the standard Gaussian. The top row of Figure 4 provides a visual check for bias, overdispersion, or underdispersion, from which users can tell that the trained model corresponding to the rightmost picture had the best out-of-sample performance among the three, and would be the best choice for constructing CONTRA regions. This is confirmed in the bottom row of Figure 4. In addition to the visual check, classical metrics can be used to quantify the deviation between the $\mathbf{z}$ sample and the standard Gaussian, aiding in model tuning."

---

> > ### Comment · Reviewer_bEdm · 2024-11-22
> > **Thank you for your answers**
> >
> > I did not doubt the validity of Theorem 1 or ask for a formal optimality proof (which may be hard in a general case). I am curious about why training an NF as in CONTRA would reduce the size of the prediction intervals. Is there an ideal case where training $t_\theta$ leads to perfect intervals?
> >
> > In Colombo, marginal validity is not broken because the NF is trained on the training (not the calibration) set. The approximation is to input conditional validity.
> >
> > Thank you for your comments on the difference between CONTRA and resCONTRA. As you said, training an NF on the residuals looks easier than training an NF for the multi-dimensional output. Why doesn't this translate into better performances? Would that be the case if the prediction model is given and does not need to be trained?
> >
> > I appreciated the details about existing multi-output conformal methods and the plots in Appendix E.
> >
> > I confirm my positive impression and will consider raising my score to 7.

---

> ### Author Response · Authors · 2024-11-22
>
> **Question 1**
>
> Thank you very much for the insightful comment! The key difference between our proposed approach (CONTRA) and STDQR is how coverage guarantees are handled. CONTRA uses Normalizing Flows to map the data into a latent Gaussian space, where the coverage guarantee is established through calibration and automatically preserved when mapping back to the original space due to bijection. In contrast, STDQR first maps the data into a latent space using a VAE and applies quantile regression. However, after transforming the points in the quantile region back to the original space, it requires a rather cumbersome calibration step to ensure coverage guarantees, which we described in section 4.1. That is, CONTRA is a simpler procedure that avoids the  adjustments in the $\mathcal{Y}$ space.
>
> **Question 2** On the question of how CONTRA and ResCONTRA compare, and if ResCONTRA always produce smoother regions.
>
> First, we have corrected our ResCONTRA procedure and rerun the data analysis. The result is that ResCONTRA is often slightly larger in size that CONTRA mostly because the new procedure requires two calibrations sets, effectively reducing the data sizes available for training. Also, ResCONTRA maps *residuals* to standard Gaussian, while CONTRA needs to map the output $\mathbf{y}$ to a Gaussian. The former is a relatively easier task because we expect residuals of any reasonable predictor to contain less information and be closer to noise than the raw outcomes $\mathbf{y}$ do. As a result, in all our examples, it took NFs with no more than two coupling layers to map the residual to a sample that resembles the Gaussian; while it takes around ten coupling layer to map the raw outcome $\mathbf{y}$ to Gaussian. And it makes sense that the simpler bijections learned in ResCONTRA lead to regions with smoother shapes.

---

> ### Author Response · Authors · 2024-11-22
>
> **Question 3** On whether the work can be extended to deal with higher-dimensional output like images.
>
> We thank the reviewer for inquiring about higher-dimensional applications. This is definitely an appealing field to explore,
> but it surely will take more time to carry out drastically different examples. Nevertheless, we read some fascinating work in the area of computing vision and identified three areas where NF are applicable and where high-dimensional prediction regions are useful.
>
>    - Object Detection: Conformal Risk Control (2022) Angelopoulos, Bates, Fisch, Lei, and Schuster.
>
> - Segmentation: Conformal Semantic Image Segmentation: Post-hoc Quantification of Predictive Uncertainty. (2024) Mossina, Dalmau, Andeol.
> -  Generation: Principal Uncertainty Quantification with Spatial Correlation for Image Restoration Problems(2023) Belhasin, Romano, Freedman, Rivlin, and Elad IEEE Pattern Analysis and Machine Learning.
>
> In addition, we’d like to cite papers that demonstrate the applicability of NF. One particularly interesting area in computing vision we encountered is Out-of-Distribution Detection (OOD). One main reason NF is a superior method than others in this area is it keeps track of the likelihood of data points, which are critical to outlier detection. We looked into the paper “Out-of-Distribution Detection for Adaptive Computer Vision” (2023) by Lind, Triebe, Nardi, and Krueger, among other papers in OOD (Iamailov, Kirichenko, Finzi, Wilson). One of the main findings of this 2023 paper is it is possible to train a normalizing flow model for OOD detection on a large and diverse dataset, namely the Microsoft COCO dataset that contains
>  million labeled instances in
>  images. They argued for training NF on features extracted from a YOLOv4 object detection network instead of on pixels to better grasp semantic information. And their choice of NF is very similar to ours: a RealNVP with
>  coupling layers, that employs two-layer neural networks of 512 hidden nodes in each of the layers.
>
> Lastly, we'd like to make an argument that, a dimension of 4 or 8
>  does not sound high, but they are interesting/hard enough in the literature of conditional density estimation and that of multi-output prediction region.  We tried a new example with dim$(Y)$=8 and got encouraging results (appended below), which shows CONTRA continues to provide the most compact prediction region with coverage guarantee.
>
>  Afterall, the ability/limitation to handle high dimension in our CONTRA is inherited from that of training conditional NF. Given there are many successful examples of NF with high dimensional variables, we are confident it can be done.
>
> -------
>
> Results below are the coverage rate and volume of various conformal regions for a real data example with dim$(Y)$=8, where each number is the average result of 20 random splits of the dataset. The standard errors (SE) for coverage rate are .003 or .004 for each method, and the SE's for volume range from 10 to 20 percent of the mean volumes.
>
> | Method         | CONTRA | PCP  | NLE  | RCP  | MCQR | Dist-split$_{bon}$ | CQR$_{bon}$|
> |----------------|--------|------|------|------|------|---------------|--------|
> | Coverage       | 0.90   | 0.90 | 0.90 | 0.91 | 0.89 | 0.90          | 0.94   |
> | Volume (*1M)   | 1.8    | 4.6  | 32   | 373  | 323  | 1007          | 1062   |
>
>
> **Question 4**
>
> We agree that the invertibility requirement in Normalizing Flows can introduce challenges, potentially making training more difficult or limiting flexibility. While we did not explore other Normalizing Flow structures in this study, invertibility is a fundamental property of all Normalizing Flows, regardless of their specific architecture. In our work, we chose the conditional RealNVP due to its simplicity and computational efficiency, which made it a practical choice for our experiments. However, our method is compatible with any Normalizing Flow structure. This flexibility allows practitioners to adopt alternative Normalizing Flows based on their specific use cases or preferences.

---

> ### Author Response · Authors · 2024-11-27
> **Thank you very much for your comments!**
>
> We appreciate the reviewer's response and new comments. Thank you for spending the time sharing your thoughts with us.
>
> **New Comment 1** : "Why training an NF as in CONTRA would reduce the size of the prediction intervals. Is there an ideal case where training leads to perfect intervals?"
>
> Great question. Yes, if the NF learned a $\mathbf{z}$ that follows perfectly standard Normal distribution, then no calibration is necessary. The 90% ball centered at the origin in the $\mathcal{Z}$ space will map back to an exact 90\% prediction region of $\mathbf{y|x}$. There is however no guarantee the 90\% prediction region is the highest probability density (HPD) region in y, but empirically, it's very close --- we checked all our simulation examples and see that the 30\% 50\%, 70\%, and 90\% conformal regions in $\mathbf{y}$ very well reflected the HPD's suggested by the actual samples from $\mathbf{y|x}$. We added pictures to **Appendix H** thanks to your question. (The pictures skipped the 30\% prediction region to avoid over crowding the picture.)
>
>
> **New comment 2**: "In Colombo, marginal validity is not broken because the NF is trained on the training (not the calibration) set. The approximation is to input conditional validity."
>
> Thanks a lot to the reviewer's comments, we get to carefully read Colombo again and now agree with what the reviewer said about their results. We updated our description of their method and their difference between us in section 1 of the draft, which is a concise version of the following:
>
> Colombo 2024 shares similarities with our ResCONTRA approach in that both aim to transform residuals into some latent z representations, of which norm serves as the most natural and effective non-conformity scores. One obvious difference is that Colombo 2024 was concerned with one-dimensional output and used relatively restricted bijections, whereas we used more flexible bijections (RealNVP) that meet the needs for multi-dimensional output.
>
> A deeper difference is the two works had different ideologies when learning the NF, though the resulting methods look rather similar: The ideal NF of Colombo is such that the resulting $\mathbf{z}$ is *independent* of $\mathbf{x}$ , and under exact independence, their prediction interval achieves conditional coverage. The base distribution of $\mathbf{z}$  is not of particular interest, but  they did use loglikelihood as the target function when learning NF and had to specify the base distribution. They tried both Normal and Uniform, and also developed a Theorem for conditional coverage probability in the particular case when $\mathbf{z}$  is learned to have approximately the Uniform distribution.
>
> Our ideal NF is such that the resulting $\mathbf{z}$  follows a *symmetric* distribution, so that the norm is the most natural non-conformity score. We also designed the base distribution to not depend on $\mathbf{x}$  so that data at different $\mathbf{x}$ can be used together for calibration. The normality is not the most important, but rather a default choice of multivariate distribution that enjoys symmetry in all directions.
>
> We did not pursue conditional coverage originally, but under a NF that can map $\mathbf{y}$  to a perfect Normal $\mathbf{z}$, we could state similar results as that of section 2.5 to Colombo. We don't feel ready to state such a result in the current paper though. For one, it's a distraction to the main contribution of moving from one-dim to multi-dim conformal. For two, we think a practical result is needed, that is, a conditional coverage result for an estimated NF like Thm 2.7 of Colombo. There is still some technical challenge as  the situations changes from a one-dimensional Uniform to a multidimensional Gaussian.
>
> We have added the following **remark** in **section 3**:
>
> **Remark:** If the distribution of $\mathbf{z}$ were perfectly independent of
> $\mathbf{x}$, Corollary 2.6 of Colombo suggests that conditional coverage probability of $\mathbf{y}_{n+1}$ given any $\mathbf{x}$ can be achieved. However, since perfect independence is not achievable in practice with finite data, they also derived in Theorem 2.7 a theoretical bound for the potential reduction in conditional coverage probabilities. This bound depends on the deviation between the learned distribution of
> $\mathbf{z}$ and the ideal base distribution that is independent of $\mathbf{x}$. Since both CONTRA and ResCONTRA aim to transform $\mathbf{z}$ to have symmetrical distributions free of $\mathbf{x}$, both methods are expected to approximately achieve the desired conditional coverage similar to those shown in Colombo. It is an on-going work to analyze the conditional coverage probabilities of CONTRA and ResCONTRA both theoretically and empirically, with the challenging but important goal of deriving practically useful bounds.

---

> ### Author Response · Authors · 2024-11-27
>
> **New comment 3** : "As you said, training an NF on the residuals looks easier than training an NF for the multi-dimensional output. Why doesn't this translate into better performances? Would that be the case if the prediction model is given and does not need to be trained?"
>
> Heuristically, when learning the conditional distribution $\mathbf{y|x}$ in two separate steps that focus on different aspects, each model can be simpler than an overall model:
>    - $\hat{f}$: estimates the mean or central tendency of $\mathbf{y}$  given $ \mathbf{x} $.
>    - $\hat{t}^*$: tries to map the residuals $ \mathbf{r}_i = \mathbf{y}_i - \hat{f}(\mathbf{x}_i)$ to a normal distribution, which help learn the variation of $\mathbf{y|x}$ around the estimated center $\hat{f}(x)$.
>
> We don't have a theory, but here are our experiences with ResCONTRA: we actually tried complicated NF (eg 10 coupling layers, same as CONTRA) for learning $\hat{t}^*$ first and found they tend to overfit the training data and lead to outlying latent $\mathbf{z}$ vectors in the calibration set. We hence used simpler NF (eg 4 or 6 coupling layers), and the latent $\mathbf{z}$ in both training and calibration can closely resemble the Normal distribution. (The good thing is that overfitting or underfitting can be diagnosed using the latent $\mathbf{z}$'s of the calibration set.)
>
> Here is our take on the issue: our core idea is to leverage the latent Gaussian $\mathbf{z}$ for calibration, and we’ve given users two options, CONTRA and ResCONTRA, to implement this idea. There will likely be scenarios where one performs better than the other, and vice versa in different contexts, and how much tuning was done for each method in each step can also influence the results. Regardless, the good news is that users don't have to pick their tool before seeing the result. They can try both methods, assess which resulting prediction region appeals to their need more (eg smaller size, or smoother boundaries). Regardless of which regions they pick, there is coverage guarantee.

---

> ### Author Response · Authors · 2024-12-03
>
> Dear Reviewer bEdm,
>
> Thank you once again for your helpful and insightful feedback. As the review process deadline approaches, we would greatly appreciate it if you could take a moment to revisit our revisions and responses to your updated comments.
>
> If you find the revisions satisfactory, we would be most grateful if you could consider raising the score to reflect the improvements.
>
> Thank you again for your support and valuable feedback.
>
> Best regards,
>
> The Authors

---

### Official Review · Reviewer_rPVJ · 2024-11-03

**Soundness:** 3
**Presentation:** 3
**Contribution:** 3
**Rating:** 6
**Confidence:** 3

**Summary:**

CONTRA is a novel method for multi-dimensional output prediction that utilizes normalizing flows to define conformal prediction regions. By mapping high-density regions in latent space to output space, it generates more precise and connected prediction regions, avoiding the oversized or irregular regions produced by traditional methods. The paper also introduces ResCONTRA, an extension that can enhance any predictive model with reliable prediction regions. Experiments demonstrate that CONTRA not only guarantees the desired coverage but also produces prediction regions with smaller volumes and smoother boundaries, outperforming existing methods on both synthetic and real-world datasets.

**Strengths:**

The manuscript is generally easy to follow and includes a sufficiently comprehensive discussion of relevant prior works. Experimental results are presented well.  The proposed solution has solid theoretical foundations.

**Weaknesses:**

- In line 186, how to construct the prediction region? \hat{E} is a q-dim ball, which contains infinitely many vectors, making its implementation unclear.
- More discussion about CNF will be valuable.
        1. Providing details about choosing q_1 in Appendix D will be better. Equal division? Moreover, for Bio dataset, the dim-y is only 1, that can not be split.
         2. A possible limitation is that normalizing flows is notoriously difficult to train in high-dimensional spaces. Extending CONTRA to handle high-dimensional outputs (e.g., image generation) would significantly enhance its practicability.

**Questions:**

None

---

> ### Author Response · Authors · 2024-11-22
>
> Thank you for your careful reading and wonderful summary! The weaknesses/questions you raised also provide us with the opportunity to improve and clarify our work. Please find our detailed responses to your inquiries below. Concise versions of these answers have been incorporated into the revised version of the paper.
>
> 1. Weakness 1: Thanks for asking us to clarify!  To construct the prediction region, we leverage the homeomorphism property of the mapping stated in Theorem 2. The theorem says that if we obtain the boundary of $\hat{E}$, rather than the entire $q$-dimensional ball in the latent space, then the image of the boundary in the $\mathcal{Y}$ space will also be the boundary of the image of the entire ball. Hence, in practice, we simply generate a large number of points uniformly distributed on the boundary of $\hat{E}$ and map these points to the $\mathcal{Y}$ space using the learned bijection. Once mapped, these boundary points in $\mathcal{Y}$ space are smoothed to form a coherent and well-defined prediction region. This approach ensures both computational efficiency and a clear implementation of the prediction region construction.
>
> 2. Weakness 2, questions about training CNF.
>
>     2.1 Please find below more details of CNF.
>
>     - $\textbf{Choosing}$   $q_1 $. For simplicity, we have set $ q_1 $ as half the dimension of $ Y $ in our experiments. This equal division balances the dimensions for the coupling layers and ensures computational efficiency. We have not implemented the following possible tricks, but they can be easily used if needed: Unbalanced splits; adopt different splits in different coupling layers.
>      - $\textbf{Training NF with one-dimensional outputs, such as the Bio dataset}.$  We'd like to remark that our CONTRA framework is compatible with any normalizing flow architecture. The coupling layer idea we emphasized is just one idea to implement the normalizing flow, and is directly applicable only to outputs with dimension 2 or higher. We list below two approaches to train normalizing flow for a one-dimensional $y$, and (a) is the one that lead to results in our tables:
>
>           (a)  Data Augmentation + Coupling layer: we added a noise output, $y_2$, to the original output, creating a two-dimensional $y$.  We train a regular RealNVP for the 2-dim $y$. When reporting the prediction region and its size, we focus on the corresponding prediction interval and its length for $y_1$.
>
>         (b) Other Normalizing Flow methods that do not use a coupling layer, such as autoregressive flows.

---

> ### Author Response · Authors · 2024-11-22
>
> 2. Weakness 2, questions about training CNF (Continued)
>
>     2.2 On the reputation of  normalizing flows for being difficult to train in high-dimensional spaces and potentially extending CONTRA to handle high-dimensional outputs such as image generation.
>
>
>      &nbsp;&nbsp;&nbsp;&nbsp;While there is existing literature on successful applications of NF in high-dimensional settings, our goal in this paper is more modest. Conformal prediction for cases with dim$(y)>$1 is highly under-explored, and we provided a strong solution for multi-dimensional problems, including successful demonstrations for dim$(y)$=$2,3,4$. Below, we added another example with dim$(y)$=8. In this way, we are already advancing the state of the art. We agree that, ideally, the method could be extended to very high-dimensional outputs, such as images.
>
>      &nbsp;&nbsp;&nbsp;&nbsp; Despite lack of time to try a very high-dimensional example in a short time, we identified three areas where NF are applicable and where high-dimensional prediction regions are useful.
>     -  Object Detection: Conformal Risk Control (2022) Angelopoulos, Bates, Fisch, Lei, and Schuster.
>     -  Segmentation: Conformal Semantic Image Segmentation: Post-hoc Quantification of Predictive Uncertainty. (2024) Mossina, Dalmau, Andeol.
>     -  Generation: Principal Uncertainty Quantification with Spatial Correlation for Image Restoration Problems (2023) Belhasin, Romano, Freedman, Rivlin, and Elad IEEE Pattern Analysis and Machine Learning.
>
>     &nbsp;&nbsp;&nbsp;&nbsp; In addition, we’d like to cite papers that demonstrate the applicability of NF. One particularly interesting area in computing vision we encountered is Out-of-Distribution Detection (OOD). One main reason NF is a superior method than others in this area is it keeps track of the likelihood of data points, which are critical to outlier detection. We looked into the paper “Out-of-Distribution Detection for Adaptive Computer Vision” (2023) by Lind, Triebe, Nardi, and Krueger, among other papers in OOD (Iamailov, Kirichenko, Finzi, Wilson). One of the main findings of this 2023 paper is it is possible to train a normalizing flow model for OOD detection on a large and diverse dataset, namely the Microsoft COCO dataset that contains million labeled instances in images. They argued for training NF on features extracted from a YOLOv4 object detection network instead of on pixels to better grasp semantic information. And their choice of NF is very similar to ours: a RealNVP with coupling layers, that employs two-layer neural networks of 512 hidden nodes in each of the layers.
>
>    &nbsp;&nbsp;&nbsp;&nbsp;   Afterall, the ability/limitation to handle high dimension in our CONTRA is inherited from that of training conditional NF. Given there are many successful examples of NF with high dimensional variables, we are confident it can be done.
>
> -------
>
>   Results below are the coverage rate and volume of various conformal regions for a real data example with dim$(y)$=8, where each number is the average result of 20 random splits of the dataset. The standard errors (SE) for coverage rate are .003 or .004 for each method, and the SE's for volume range from 10 to 20 percent of the mean volumes.
>
> | Method         | CONTRA | PCP  | NLE  | RCP  | MCQR | Dist-split$_{bon}$ | CQR$_{bon}$ |
> |----------------|--------|------|------|------|------|---------------|--------|
> | Coverage       | 0.90   | 0.90 | 0.90 | 0.91 | 0.89 | 0.90          | 0.94   |
> | Volume (*1M)   | 1.8    | 4.6  | 32   | 373  | 323  | 1007          | 1062   |

---

> > ### Comment · Reviewer_rPVJ · 2024-12-02
> >
> > Thank you for addressing most of the concerns. I maintain my recommendation to accept this paper.

---

### Official Review · Reviewer_e6SY · 2024-11-04

**Soundness:** 3
**Presentation:** 4
**Contribution:** 3
**Rating:** 8
**Confidence:** 2

**Summary:**

This work addresses the limitation of most of the current conformal prediction algorithms which struggle with multi-dimensional outputs. The proposed method CONTRA, uses conditonal normalizing flows (CNFs) to learn a bijection, mapping a latent variable form a simple base distribution to the output variable, and then mapping the high density regions of the base distribution to the output. Using Conformal Prediction ( particularly the split conformal prediction algorithm) , CONTRA then calibrates the results to provide marignal coverage gaurantees of CP on including the true prediction region.

**Strengths:**

1. The writing is clear, and to the point. The contributions are presented honestly and placed well with respect to the prior work.
2. The experiments are elaborate, and show the main metrics in CP: efficiency in terms of the size ( volume ) of the prediction sets and coverage. Both synthetic and real world datasets are considered, and the results are consistent across the two settings ( synthetic vs real-world).
3. the output regions of CONTRA do seem to visually support the authors' claims in that they are interpretable and have smoother boundaries compared to the related work. In most settings/datasets CONTRA seems to be outperforming most other baselines ( except PCP ).
4. They address an important and practical problem (multi-modal outputs ) which extends the applicability of Conformal Prediction.
5. Their methods seems simple to follow and implement.
6. They introduce variation of the method, to adopt to different choices of the fitted model for the conditional density estimation.

**Weaknesses:**

1. PCP ~ ( a competing method) is able to outperform both variations of CONTRA and ResCONTRA in some cases in terms of providing smaller prediction regions while maintaining coverage guarantees.
2. the authors mention instances in which CONTRA could deteriorate in performance. Particularly when its hard to learn bijection well. but no results are provided to illicit the deterioration trends or to show the remedy solutions discussed by the authors can help.

**Questions:**

1. with respect to point 2 in the weaknesses, how do competing methods, particularly PCP, are expected to perform in cases that CONTRA is not able to perform well. A discussion ~ potentially a brief experiment on synthetic data ~ where we understand the trend of performance deterioration in CONTRA, and the effectiveness of the solutions proposed on page 10,  could illicit a more clear choice to follow in whether one should try to apply the proposed solutions or go with another method.

---

> ### Author Response · Authors · 2024-11-22
>
> We sincerely appreciate your time and expertise. Your summary, along with the identification of strengths and weaknesses, is spot on, and we’re truly grateful for your thoughtful and thorough review! We will address the weaknesses and questions below.
>
> - Weakness 1: Thank you very much for the comment! We agree that PCP shows strong performance in certain cases by producing smaller prediction regions while maintaining coverage guarantees. Note that the small volume of PCP is partly achieved by allowing the prediction region to be unions of thousands of balls. The fact that CONTRA employs a much smoother and interpretable region and still consistently attains similar and sometimes smaller volume is impressive. That is, if we develop a more comprehensive criteria of *good* prediction region that takes into account small volume and smoothness, CONTRA would easily become the preferred choice.
>
>
> -  Weakness 2 and Question 1 on how CONTRA and competitors compare in situations where NF is challenging to learn.  We did more experiment when (1) the data size is too small for NF to be well-trained, as well as when (2) the NF model structure is intentionally made very simple and fails to fit the training and calibration data well. We believe these experiments and their results shown in the new Appendices E and F shed light on what happens when NF is challenging to learn.
> We quote a paragraph from Appendices E "A straightforward guideline to mitigate these overfitting and underfitting issues is to evaluate how closely the latent variables $\mathbf{z}$ from the calibration set resemble a random sample from the standard Gaussian. The top row of Figure 4 provides a visual check for bias, overdispersion, or underdispersion, from which users can tell that the trained model corresponding to the rightmost picture had the best out-of-sample performance among the three, and would be the best choice for constructing CONTRA regions. This is confirmed in the bottom row of Figure 4. In addition to the visual check, classical metrics can be used to quantify the deviation between the $\mathbf{z}$ sample and the standard Gaussian, aiding in model tuning."

---

> > ### Comment · Reviewer_e6SY · 2024-12-02
> >
> > Thank you for your comments addressing the weaknesses, and on including additional experiments addressing point 2.
> >
> > I maintain my score and confidence level.

---

### Official Review · Reviewer_vtsj · 2024-11-04

**Soundness:** 3
**Presentation:** 3
**Contribution:** 2
**Rating:** 5
**Confidence:** 4

**Summary:**

The authors propose a new conformal prediction method for multi-dimensional regression. It is based on fitting a (conditional) normalizing flow in the space of outputs. Authors perform numerical experiments on public datasets that illustrate certain benefits of the proposed method.

**Strengths:**

- The proposed method makes a lot of sense as flexible estimator of conditional label distribution is obviously required to achieve optimal set size, especially in multidimensional case.
- The results are adequately explained.
- Experimental results show certain benefits of the proposed approach in terms of prediction set size.

**Weaknesses:**

- The proposed approach boils down to application of the normalizing flows to the split conformal prediction framework. There are no technical difficulties on this way. Any other conditional density estimator can be used here instead.

- All the theorems in the paper are standard (not sure, why even proofs are provided). There seems to be no theoretical contribution for the present paper.

**Questions:**

- Why don't you study other density estimation methods?

- Performance of normalizing flows heavily depends on the size of the dataset. How will it influence the results of the proposed method? I think that this should be studied.

---

> ### Author Response · Authors · 2024-11-22
>
> **Weakness 1**
>
> We respectfully disagree with the weakness statement from the reviewer that "any other conditional density estimate can be used here".  First of all, the specific procedure we proposed as our main contribution, CONTRA, deliberately takes advantage of the latent representation $\mathbf{z}$ in a normalizing flow (NF) model and can not be applied to methods other than NF. And the main point of our paper is we demonstrated the unique advantage of doing so in the conformal prediction framework, which is a novel and intentional choice. Secondly, in the existing literature, the only other types of multi-dimensional density estimation methods that have been used to construct conformal regions are Generative models, such as VAE. And the tool that add conformal regions to them are the PCP and ST-DQR. Both methods have been described in details in section 4 in our paper.
>
> We do now included more comparisons of our method to PCP and ST-DQR, where we implemented diffusion model + PCP,  as well as diffusion model + ST-DQR. (Note that the combination of diffusion model and conformal prediction are also quite new, and we have not seen people done it, though the ST-DQR paper did mention this combination.) Please see Figure 1 for the results. Our conclusion remains the same as in the previous draft that CONTRA and ResCONTRA are able to achieve similar sized regions that are much more connected and smoother than that of these state-of-the-art competitors.
>
> We want to mention that the invertible mapping of the output $\mathbf{y}$ to a Normal latent variable $\mathbf{z}$ is exactly what makes the calibration based on the norm of $\mathbf{z}$ a well-suited procedure. Simply put, one would not draw a perfect ball around the highest density point of any probability density and call that a high density region, but one could comfortably say that for a standard multivariate normal distribution.
>
> **Weakness 2**
>
> Conformal prediction is a very general framework that ensures marginal coverage guarantees for the resulting prediction regions. However, its success depends solely on the implementation details--- "the devil is in the details."  One most important decision to make when operating within the framework is to pick a definition of the non-conformity score $s(y, h(\mathbf{x}))$, where $h(\mathbf{x})$ denotes any point or region estimate (such as those resulted from any density/quantile estimator) for $\mathbf{y}$. This choice of $s$ is CRITICAL to the *properties of the prediction region* and the subject of many new advancements in the conformal prediction literature. This is particularly under-studied for the multi-dimensional y case, and exactly our topic.
>
> Two major *properties of a prediction region* are their volume and their adaptivity to $\mathbf{x}$. For example, the naive choice of $s$ as the Euclidean distance when $h$ is a point estimator leads to conformal regions that are balls with the SAME radius centered at $h(\mathbf{x})$ at EVERY $\mathbf{x}$. Leading to NO Adaptivity, restricted shapes, and consequently large sizes. For another example, in the one-dimensional $\mathbf{y}$ case when $h$ is a pair of quantiles, the well-known solution, conformal quantile regression (CQR) by Romano, Patterson and Candes (2019) uses a definition of $s$ that eventually gives the same amount of margin correction to $h(\mathbf{x})$. That is, CQR allows adaptivity to $\mathbf{x}$ inherited from $h(\mathbf{x})$, but the correction provided by the calibration step brings in no additional adaptivity, leaving room for improvement. This is exactly where our method comes in. Our idea of improvement not only makes the correction more adaptive, but also extends the range of application from one to multi-dimensional $\mathbf{y}$. We firmly believe our proposal is a novel and solid contribution to the literature that many users and researchers would appreciate.
>
> We greatly appreciate the reviewer's comments and questions, which showed us that we may not have fully clarified the essence of our idea or did enough comparisons to competitors in the previous draft. In response, we have rewritten part of Section 3 to enhance the clarity of our presentation and included more examples (Figure 1 and 5) and we hope the new draft leaves no doubt regarding the contribution of our work.

---

> > ### Author Response · Authors · 2024-11-22
> >
> > **Question 1**.
> >
> > From our response to weaknesses 1 and 2, we have partially illustrated why we focus on studying conformal prediction based on normalizing flows. We now provide additional details to show advantages of our CONTRA. For the taxi data with 4800 samples, the training stage of the diffusion model took 15 seconds, shorter than the 65 seconds for NF. During the calibration stage of PCP/STDQR, $K$ points needs to be simulated for each $\mathbf{x}$. E.g., when $K = 3000$, the diffusion model took 20 minutes to calibrate, while the normalizing flow model took 5 seconds. In contrast, using CONTRA only requires one point estimator for the inverse process for each $(\mathbf{x}, \mathbf{y})$, and calibration took less than 1 second.  After all, the proposed CONTRA took 65+1=66 seconds, the NF + STDQR (or PCP) took 65+5=70 seconds, while the diffusion model + STDQR (or PCP) took 20 minutes and 15 seconds. Showing that our CONTRA methods has a great advantage in computing cost.
> >
> > The resulting prediction regions of both competitors look reasonable in their coverage, but again, each region is the union of thousands of disks, leading to highly complicated shapes. We think our CONTRA region is much simpler and no bigger in size, yet are faster to obtain. We thank the reviewer for suggesting us to try harder running competing methods, which further demonstrated the elegance and value of CONTRA.
> >
> > **Question 2** concerning the impact of sample size on the performance of NF.
> >
> > This is a great suggestion. We have now studied the impact of different sample size  on both CONTRA and its strongest competitor "diffusion model + ST-DQR" using the Taxi data. The results are reported in Appendix F. Figure 5 shows the prediction regions when we change the training+calibration size (from 300, 1200, to 4800). And we can see the NF did improve as sample sizes increases, and stabilizes with larger data sizes.
> > Nevertheless, the beauty of the conformal method is that regardless of the quality of the NF, the resulting conformal region has guaranteed coverage  --- in case the base prediction method (such as the NF) was not well learned, conformal prediction regions will automatically be large thanks to the calibration step (as in column (a) of Figure 5). we display the distribution of the latent $\mathbf{z}$ learned by NF of the calibration dataset. Closer resemblance of the $\mathbf{z}$'s to the Bivariate Normal distribution suggest better training of the NF.

---

> ### Comment · Reviewer_vtsj · 2024-11-26
> **Further comments**
>
> Dear authors,
>
> Thanks for your detailed answers. I appreciate your comment on Weakness 1. I now see the value of using normalizing flow due to the trick with the norm. That justifies the usage of normalizing flows, while usage of other methods seem to be indeed complicated.
>
> On the other hand, I am not satisfied with your comment on the Weakness 2. My point was that I don't see much value in Theorems 1-3 as they are all standard theorems either from conformal prediction literature or from mathematical analysis. Your comments about adaptivity are well understood but the question was not about the adaptivity at all.
>
> I appreciate the answers on questions 1 and 2 and additional experiments performed. I think that the experiment with increasing sample size should be extended by studying the coverage, standard deviation of coverage and confidence set volume. Currently this experiment looks very informal.
>
> Finally, the point that I missed during initial review but find important is study of conditional coverage. This is the central point of many recent papers on conformal prediction including PCP paper. That's very interesting to see what are the properties of CONTRA compared to the competitors.
>
> Overall, based on the rebuttal, I increase my score to 5. The reason for the increase is a nice idea on finding highest density regions for normalizing flows. The reasons to not increase further:
> 1. The theory in the paper doesn't seem relevant and new.
> 2. Experimental part will benefit from the study of conditional coverage which is important for all the adaptive conformal methods.

---

> > ### Author Response · Authors · 2024-11-27
> > **Thank you for your comments.**
> >
> > We appreciate the reviewer's response and new comments. Thank you for spending the time sharing your thoughts with us.
> >
> > **New Comment 1**: About the concern of Weakness 2.
> >
> > We are filling a practical gap of using conformal in multi-dimensional outputs. Our main contribution is in providing a couple of new ideas and demonstrating their power and potentials. Proving new theoretical results is not our goal.
> >
> > We stated some results as theorems because they are a good way to organize the materials, and it's hard to simply cite other sources because these results were not stated or proved elsewhere in a way that fits in our current context. We have no intention to make our readers think the results are entirely new. As a result, we decide to rename the *theorems* as
> >  *proposition*, as well as add an extra line explaining they are simple corollary of existing results and/or which books contain similar materials.
> >
> > **New Comment 2**: "I think that the experiment with increasing sample size should be extended by studying the coverage, standard deviation of coverage and confidence set volume. Currently this experiment looks very informal."
> >
> > That is a good point. Repetitions and assessing variability is important and the reason we did replications for all examples in the main text and provided averages and standard errors of coverage and region sizes. We have not yet had the time to do the same for all the new examples added to the Appendices, but such replications are very much do-able in the coming weeks. The current pictures are very informative though, that they show the amount of change in shape of the prediction regions, which could be more interesting for practitioners than numerical summaries for the multi-dimensional outputs.
> >
> > **New Comment 3**: "Finally, the point that I missed during initial review but find important is study of conditional coverage. This is the central point of many recent papers on conformal prediction including PCP paper. That's very interesting to see what are the properties of CONTRA compared to the competitors."
> >
> > This is a very good point. We mentioned our take on conditional coverage probability in our reply to another reviewer, bEdm. Below is one paragraph from that reply that is most relevant to your request:
> >
> > We did not pursue conditional coverage originally, but under a NF that can map $\mathbf{y}$  to a perfect Normal $\mathbf{z}$, we could state similar results as that of section 2.5 to Colombo. We don't feel ready to state such a result in the current paper though. For one, it's a distraction to the main contribution of moving from one-dim to multi-dim conformal. For two, we think a practical result is needed, that is, a conditional coverage result for an estimated NF like Thm 2.7 of Colombo. There is still some technical challenge as  the situations changes from a one-dimensional Uniform to a multidimensional Gaussian.
> >
> > We have added the following **remark** in **section 3**:
> >
> > **Remark:** If the distribution of $\mathbf{z}$ were perfectly independent of
> > $\mathbf{x}$, Corollary~2.6 of Colombo suggests that conditional coverage probability of $\mathbf{y}_{n+1}$ given any $\mathbf{x}$ can be achieved. However, since perfect independence is not achievable in practice with finite data, they also derived in Theorem 2.7 a theoretical bound for the potential reduction in conditional coverage probabilities. This bound depends on the deviation between the learned distribution of
> > $\mathbf{z}$ and the ideal base distribution that is independent of $\mathbf{x}$. Since both CONTRA and ResCONTRA aim to transform $\mathbf{z}$ to have symmetrical distributions free of $\mathbf{x}$, both methods are expected to approximately achieve the desired conditional coverage similar to those shown in Colombo. It is an on-going work to analyze the conditional coverage probabilities of CONTRA and ResCONTRA both theoretically and empirically, with the challenging but important goal of deriving practically useful bounds.

---

### Author Response · Authors · 2024-11-22

Thank you very much for your valuable suggestions and insightful questions. Based on your feedback, we have revised our draft, highlighted the updated sections in blue, and carefully replied to all the questions. These revisions and responses address concerns regarding the choice of normalizing flows for conformal prediction, additional comparisons with competing methods (e.g., diffusion models with PCP and ST-DQR), more details about normalizing flows (NFs), the novelty of ResCONTRA, and discussions on issues of underfitting/overfitting, strategies to avoid them, and performance with small training sizes.

---

### Meta-Review · Area_Chair_qfSu · 2024-12-19

**Metareview:**

The main claim of the paper is that conformal prediction with multidimensional outputs can be effectively handled using a new proposed method called CONTRA.

The method is simple to implement and outperforms most baselines, and addresses an important and practical problem in CP.  Therefore, it is of interest to the wider CP community attending ICLR.

The main weakness of the work is that the theoretical results (Theorems 1-3) are existing standard results, and the proposed method is a rather straightforward applications of normalizing flows to conformal prediction.

I encourage the authors to take the reviewer's feedback into account for the final version of the paper.

**Additional Comments On Reviewer Discussion:**

All reviewers gave high scores and like the contribution of the work. Reviewer vtsj pointed out that the method is a straightforward application of normalizing flows to CP and that the theorems in the paper are known standard results. The first issues was cleared up during the rebuttal, whereas the second one still stands. However, the empirical results and the fact that the method addresses an important problem in a clean and simple fashion are enough in my opinion to accept the work as a poster at ICLR.

---

### Decision · Program_Chairs · 2025-01-22

Accept (Poster)